# First characterization of a new perturbation system for gust generation: The Chopper

Ingrid Neunaber[1] and Caroline Braud[1]

[1]LHEEA - Ecole Centrale Nantes, CNRS, 1 Rue de la Noë, 44321 Nantes, France

**Correspondence:** Ingrid Neunaber (ingrid.neunaber@ec-nantes.fr)

**Abstract.** We present a new system for the generation of rapid, strong flow disturbances in a wind tunnel that was recently installed at Ecole Centrale Nantes. The system is called the *chopper*, and it consists of a rotating bar cutting through the inlet of a wind tunnel test section, thus generating an inverse gust that travels downstream. The flow generated by the chopper is investigated with respect to the rotational frequency using an array equipped with hot-wires that is traversed downstream in the flow field. It is found that the gust can be described as a superposition of the mean gust velocity, an underlying gust shape and additional turbulence. Following this approach, the evolution of the mean gust velocity and turbulence intensity are presented, and the evolution of the underlying inverse gust shape is explained. The turbulence is shown to be characterized by an integral length scale of approximately half the chopper blade width and a turbulence decay according to $E(f) \propto f^{-5/3}$.

## 1 Introduction

The atmospheric boundary layer is naturally turbulent, and these turbulence conditions are highly complex and non-stationary. One characteristic of atmospheric turbulence is the significantly higher probability of extreme events, i.e. gusts, compared to a Gaussian distribution that is called *intermittency* (see e.g. Morales et al. (2011)). Gusts can occur in stream-wise direction (see e.g. Bardal and Sætran (2016) and Letson et al. (2019)), but also in transverse direction, for example in form of thunderstorm downbursts (see e.g. Nguyen and Manuel (2014)), or in form of sudden wind direction changes. This shows that the working conditions in this part of the atmosphere are challenging, and in different fields of application, research is carried out to optimize performances, for example in wind energy research but also in unmanned aerial vehicle research.

Wind energy converters are operating in the atmospheric boundary layer, and one consequence of the present turbulence is that the wind turbine experiences higher loads, also due to the intermittency in the wind (see e.g. Frandsen (2007), Lee et al. (2012) and Schwarz et al. (2019)). One reason is that a rapid change of the velocity can not be accounted for with today's control mechanisms. To still ensure the durability and safety of a wind turbine during its planned lifetime of generally 20 years, the IEC-61400-1 norm gives a set of design requirements (cf. IEC-61400-1-4 (2019)) that include the turbulence characteristics and the occurrence of extreme wind conditions. One of the extreme wind conditions is the extreme operating gust (EOG), an artificial gust that is assumed to occur once every 50 years. It is modeled as a Mexican hat wavelet and characterized by a fixed characteristic time $\Delta t_{IEC} = 10.5\,\text{s}$, a rise and fall time $t_r = t_f = 2.8\,\text{s}$, and a velocity amplitude $\Delta u$[1]. $\Delta u$ can be in the

---

[1]The velocity amplitude $\Delta u$ depends on the velocity $u$ at hub height $z_{hub}$, the rotor diameter $D$ and the turbine class.

order of magnitude of $10\,\mathrm{ms^{-1}}$ for an inflow velocity close to the turbine's cut-off velocity[2]. One particular consequence of these strong and rapid gusts is that the aerodynamics at the rotor blade are strongly affected, and the interactions between the three-dimensional, turbulent inflow and the rotor blades are highly complex and not well understood.

To investigate the behavior of wind turbines and the aerodynamic reaction of rotor blades to turbulence and gusts, laboratory scale experimental research is an acknowledged method. Here, the inclusion of turbulence and unsteady flow conditions is important to model the real atmospheric conditions more closely. Over the years, different approaches have been used to include turbulence in wind tunnel experiments, and they can be classified into two categories: passive and active.

Among passive approaches, grids are often used in the wind tunnel to include turbulence in airfoil and rotor aerodynamics investigations, as for example done by Devinant et al. (2002), Sicot et al. (2006) and Sicot et al. (2008) for airfoils and Bartl et al. (2018) for investigations of the wake of a lab scale turbine. For the investigation of for example model wind turbines, another approach is to model the atmospheric boundary layer by means of static elements, as discussed e.g. by Lee et al. (2004) and Hancock and Hayden (2018) and used for example by Aubrun et al. (2013), Bastankhah and Porté-Agel (2017) and Chamorro et al. (2012). Here, mean and turbulent velocity profiles of neutral or weakly stable atmospheric flows are often established by a long series of trial and error tests using passive devices such as spikes, crossbars, and/or roughness elements. While these setups give important information on how turbulence influences the result, they are limited to stationary investigations. Also, as for example Bartl et al. (2018) write, "the inevitable decay of the grid-generated turbulence in the experiment is not representative for real conditions ".

To overcome these limitations, setups that actively control the inflow in the wind tunnel can be used.

One example for this is the generation of transverse gusts by inducing flow perpendicular to the usual stream-wise direction, as done for example by Corkery et al. in a water channel or Poudel et al. (2018) in a wind tunnel to investigate the response of a flat plate respectively airfoil to a vertical flow. While in the former two examples, an inlet and an outlet exist for the transverse gust to pass through, Zhang et al. (2015) simulate a thunderstorm downburst similarly by blowing air down onto the test section floor.

To generate stream-wise gusts in the test section, the flow can globally be altered in the wind tunnel. Horlock (1974) use flexible test section walls in a "wavy-wall" wind tunnel that can be driven to produce sinusoidal transverse gusts and stream-wise gusts. In Sarkar and Haan, jr. (2008), a system of bypass ducts is installed in a closed-loop wind tunnel to change the flow, and they achieve a 27% velocity increase in $2.2\,\mathrm{s}$. Greenblatt (2016) uses a blockage modulation at the end of the test section of an open jet facility to induce flow variations. As these approaches focus on the flow variation, turbulence is not added to the gusts. Another approach to generate stream-wise gusts is to alter the flow directly at the beginning of the test section with active elements. The methods depend on the aimed gust shape. One possibility is the generation of periodic gusts in the wind tunnel. Tang et al. (1996) use rotating slotted cylinders to generate periodic gusts in the lateral and longitudinal directions, and they are able to create sinusoidal velocity fluctuations with an amplitude of $\Delta u = 5\,\mathrm{ms^{-1}}$ at an inflow velocity of $u = 23\,\mathrm{ms^{-1}}$. While the velocity amplitude is only half of the presented EOG's velocity amplitude at $u = 25\,\mathrm{ms^{-1}}$, one advantage of this system is

---

[2]For example, the EOG used for a wind turbine of class I.A with diameter $D = 136\,\mathrm{m}$ and hub height $z_{hub} = 155\,\mathrm{m}$ has an amplitude of $\Delta u = 9.8\,\mathrm{ms^{-1}}$ at $u = 25\,\mathrm{ms^{-1}}$.

the simple, low-cost design. Another method is to use a single pitching and plunging airfoil, as Wei et al. (2019a) demonstrate, who aim for a low-turbulent, lateral, periodic gust. Wester et al. (2018) and Wei et al. (2019b) use an array of oscillating 2D airfoils at the inlet of the wind tunnel. Wester et al. (2018) generate a lateral gust similar to the artificial EOG proposed in the IEC-61400-1 norm while Wei et al. (2019b) intend to reproduce an "ideal" sinusoidal gust without turbulence in lateral and longitudinal direction.

Another device that is capable of creating customized flows and also gusts is an active grid. The most common design was proposed by Makita (1991), the so-called 'Makita-style' active grid, initially to generate homogeneous, quasi-isotropic turbulence in small wind tunnels. This initial design has been modified by different groups over the past years (e.g. Bodenschatz et al. (2014) and Hearst and Lavoie (2015)), and different control strategies have been applied - not only to push turbulence research but also to use customizable flows for wind tunnel experiments. Cekli and van de Water (2010) and Hearst and Ganapathisubramani (2017) generate sheared flow profiles. Wind fields with atmospheric-like statistics are generated in Knebel et al. (2011). Traphan et al. (2018) and Petrović et al. (2019) demonstrate that Makita-style active grids can be used to generate gusts matching the one proposed as EOG in IEC-61400-1-4 (2019). While the flexibility and the stage of research are two major advantages of the Makita-style active grid, disadvantages include high complexity, high cost and maintenance. Also, it should be emphasized that active grids struggle to create sudden, strong changes such as those that can be made with the here presented device.

Overall, different systems to generate gusts and specific flows exist. It was demonstrated by the progress in active grid research and atmospheric wind tunnel facility flows that such perturbation systems need time to be fully customized. Also, many of the above-presented experiments focus on the generation of specific gusts and flows tailored to investigate isolated aerodynamic effects.

In this study, we will present a new perturbation system with a unique mechanism consisting of a rotating device. The objective of this new perturbation system, called the "chopper", is to combine sudden, strong mean flow changes and a turbulent background which is normally targeted separately in other perturbation devices. Targeted are the perturbations that cannot be accounted for with today's global control mechanisms and that will thus strongly affect the blade and rotor aerodynamics. This setup will therefore help to improve the insight into blade and rotor aerodynamics. Moreover, its application is also particularly interesting for the development of active flow control (ACF) devices. Using for example plasma actuators, a fast and local control with up to $10\,kHz$ can be achieved, and up to a few kHz can be achieved in case of micro-jet actuators (see e.g. Leroy et al. (2016) and Jaunet and Braud (2018)). Those local flow control mechanism systems are suitable to alleviate loads from rapid and turbulent atmospheric perturbations and thus decrease fatigue loads.

The present paper is a proof of concept of this new system, and it focuses on investigating the mean flow behavior, the shape and the characteristic time of the generated perturbation and the turbulence generated by the system throughout the test section. The amplitude and the characteristic time will be compared to the IEC EOG. Further customization of the perturbation system will be needed to match realistic gusts like the one discussed for example by Bardal and Sætran (2016) which are far from the ideal gust shape provided by the IEC-61400-1 norm. Therefore, another objective of this proof-of-concept article is to propose some guidelines towards that direction.

The paper is organized as follows: In section 2, the experimental setup will be introduced including a description of the chopper, section 3 will present results including the analysis of the time series (section 3.1), the mean gust flow field (section 3.2), the underlying shape (section 3.3) and the gust fluctuations (section 3.4), and section 4 will conclude the paper with an outlook.

## 2 Experimental setup

In the following, the chopper and the experimental setup used to characterize the flow disturbance generated by the chopper are introduced. In figure 1, a sketch of the chopper and its installation in the closed-loop wind tunnel can be found. Due to the installation of the chopper, a gap in the inlet causes a discontinuity upstream in the test section. The chopper consists of a straight rectangular bar, the *chopper blade*, that rotates around its central axis which is aligned parallel to the test section. Within one revolution, the two arms of the chopper blade cross the inlet of the test section. This configuration induces a turbulent shear flow in the wall-normal (i.e. Z) direction, perpendicular to the stream-wise inflow. Also, because the blade tip is moving faster than the blade root, a gradient is expected in the transverse direction. In preference to designing a complex blade shape to counteract this effect, we kept the shape as simple as possible because we expect that far downstream the flow will evolve towards a homogeneous flow in the XY-plane. With this design, rapid and strong velocity variations can be generated while turbulence is also included. In addition, this setup can be customized easily in the future by changing the chopper blade. The chopper frequency $f_{CH}$ is defined as the frequency with which the blade arms cross the inlet, making it twice the rotational frequency, $f_{rot} = 1/2 \cdot f_{CH}$. When the blade's arms cut through the inlet of the wind tunnel, an inverse gust is created and transported downstream in the test section. The current chopper blade has a radius of $R_{CH} = 900\,\text{mm}$, a width of $w_{CH} = 200\,\text{mm}$, and a thickness of $d_{CH} = 50\,\text{mm}$.

The test section's dimensions are $(500 \times 500)\,\text{mm}^2$ with a closed test section of $2300\,\text{mm}$ length. The turbulence intensity of the wind tunnel is below 0.3% throughout the test section, and it was verified prior to the experiments that the flow in the empty wind tunnel is homogeneous. In the following, the measurement positions will be given with respect to the chopper blade, and the vertical and lateral positions will be given with respect to the centerline of the test section. The coordinates will be normalized by the blade width. The blockage induced by the chopper varies with the angle $\beta$ between the blade and the horizontal centerline of the inlet and can reach up to 40.6%. An induction sensor gives a signal every time one of the blade arms passes the horizontal centerline of the inlet ($\beta = 0$). From this measurement, the blockage (i.e. area of the chopper blade inside the inlet versus the total area of the inlet) is computed.

To characterize the chopper, an array equipped with five 1D hot-wire probes of type 55P11 with a sensor length of $1.25\,\text{mm}$ from Dantec Dynamics was used. A sketch of the probe arrangement can be found in figure 1. Four of the sensors were operated by MiniCTA modules of type 54T30 from Dantec Dynamics with a hard-ware low-pass filter set to $f_{LP} = 10\,\text{kHz}$ while one sensor, the sensor positioned at the centerline, was operated using a Disa Type 55M01 Main Unit and Standard Bridge, and the temporal resolution of the electronics was determined with a square-wave test to be $38\,\text{kHz}$. A temperature probe was used to monitor the temperature during calibration and measurement, and a temperature correction according to Hultmark and Smits (2010) was applied when processing the data. The hot-wires were calibrated in the range $0\,\text{ms}^{-1} \leq u \leq 55\,\text{ms}^{-1}$ using a Disa

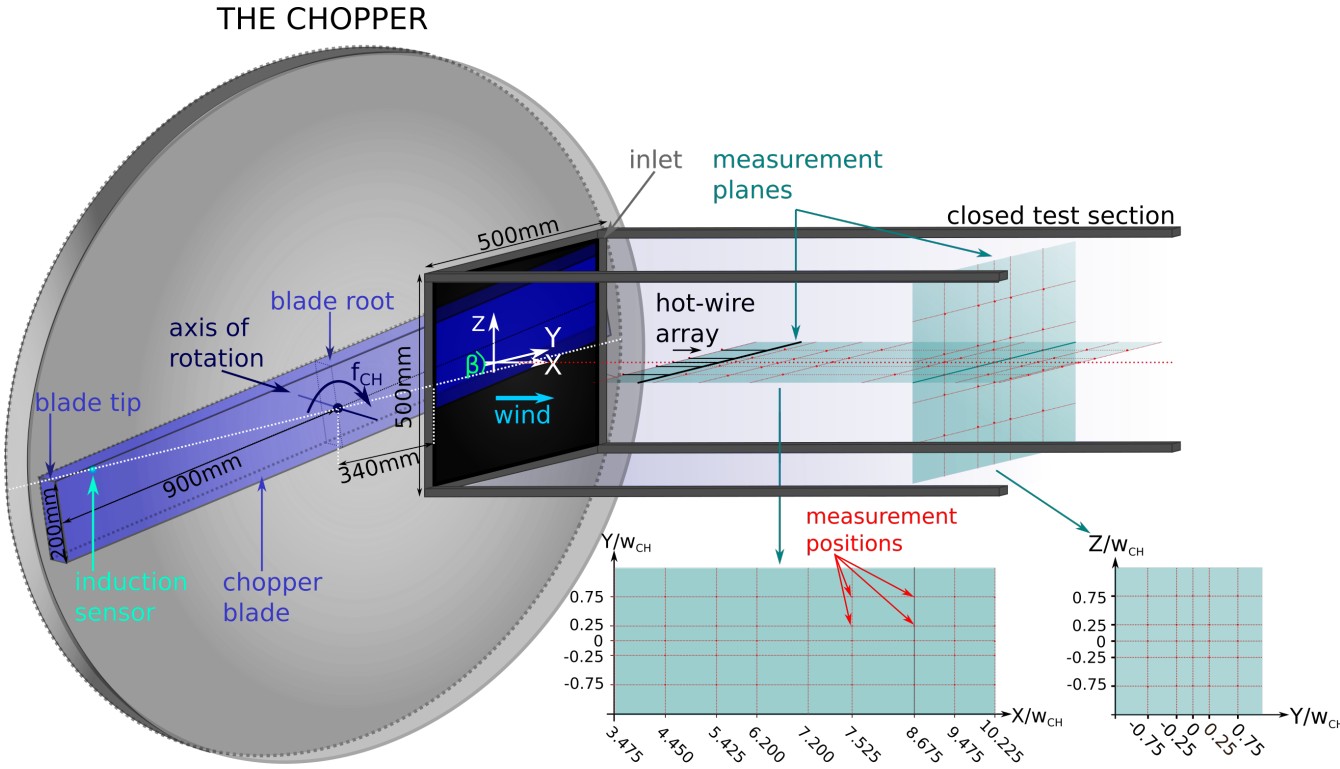

**Figure 1.** Experimental setup: The rotating chopper blade (blue) cuts through the inlet of the wind tunnel test section, thus generating a flow disturbance. An induction sensor returns a signal when the chopper blade crosses the horizontal centerline of the test section inlet. Measurements were carried out using an array of five hot-wire probes at the positions marked with red dots within the turquoise measurement planes.

calibrator of type 55D45 with a nozzle of $60\,\mathrm{mm}^2$. The data was collected by a D.T.MUX Recorder with a sampling frequency of $f_s = 100\,\mathrm{kHz}$, and the data was filtered with a built-in low-pass hard-ware filter at $f_{LP} = 40\,\mathrm{kHz}$. We are aware that the wake of the rotating chopper blade is complex and three-dimensional which may limit the use of 1D hot-wire anemometry. However, the scope of this study is the investigation of the evolution of the mean velocity, inverse gust shape and the turbulence characteristics throughout the test section. As we scan the flow field with several probes and at multiple positions, arriving at 65 measurement points in total, a spatial three-dimensional characterization at a very high sampling frequency is achieved that allows us the detailed investigation of these quantities and also the rapid flow reactions and the high frequent, small-scale turbulence.

Horizontal velocity profiles were measured at nine downstream positions at the test section's centerline, and in addition, one plane perpendicular to the flow at $X = 8.675\,w_{CH}$ was scrutinized. The measurement planes are indicated in figure 1. The inflow velocity was $u_0 = 25\,\mathrm{ms}^{-1}$, the velocity we will use for aerodynamic measurements of an airfoil in the future. Two chopper frequencies, $f_{CH1} = 0.04\,\mathrm{Hz}$ and $f_{CH2} = 0.4\,\mathrm{Hz}$, have been investigated. The frequencies were chosen to generate

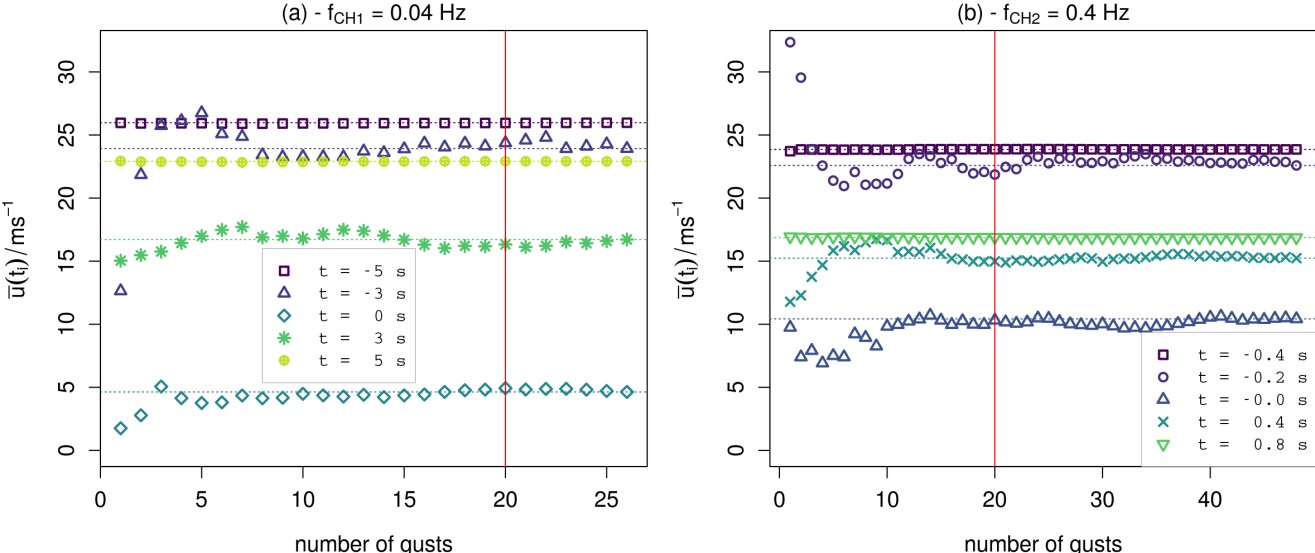

**Figure 2.** Convergence test of the phase average at five different points of time during the gust event for a data set measured at the centerline at $X = 3.475 w_{CH}$ and the two chopper frequencies $f_{CH1}$ (a) and $f_{CH2}$ (b).

a) inverse gusts with a characteristic time similar to the gust characteristic time $\Delta t_{IEC} = 10.5 \, \text{s}$ used in the IEC-61400-1 norm, and b) inverse gusts with a characteristic time that is scaled down to wind tunnel dimensions. For this, the ratio of gust characteristic time over chord length $\Delta t/c$ is kept constant, which yields $\Delta t \approx 1 \, \text{s}$ for a blade chord length of $c = 0.1 \, \text{m}$ and thus $\approx 1 \, \text{s}/0.1 \, \text{m}$ in the wind tunnel as compared to $\approx 10 \, \text{s}/1 \, \text{m}$ for full scale measurements. To measure a sufficient amount of inverse gust events for each chopper frequency, the measurement times were adapted and are $t_1 = 600 \, \text{s}$ for $f_{CH1} = 0.04 \, \text{Hz}$ and $t_2 = 120 \, \text{s}$ for $f_{CH2} = 0.4 \, \text{Hz}$, respectively. It was verified that the number of inverse gusts captured is sufficient to reach statistical convergence, which is achieved when capturing at least 20 inverse gust events as illustrated exemplarily in figure 2 for various points of time $t$ during the respective gust event.

In the next section, the results from these measurements are presented.

## 3 Results

In the following, the structure and the downstream evolution of the inverse gust generated by the chopper will be presented. For a first overview, an example of the time series will be shown in combination with the energy spectral density. The separate elements of the gust structure will be explained. Afterwards, the evolution of the gust will be investigated by looking at the average gust flow field, the inverse gust shape and the turbulence within the gust. The analysis will include mean velocities $u$, turbulence intensities $TI$ and a spectral analysis including the computation of the integral length scale $L$.

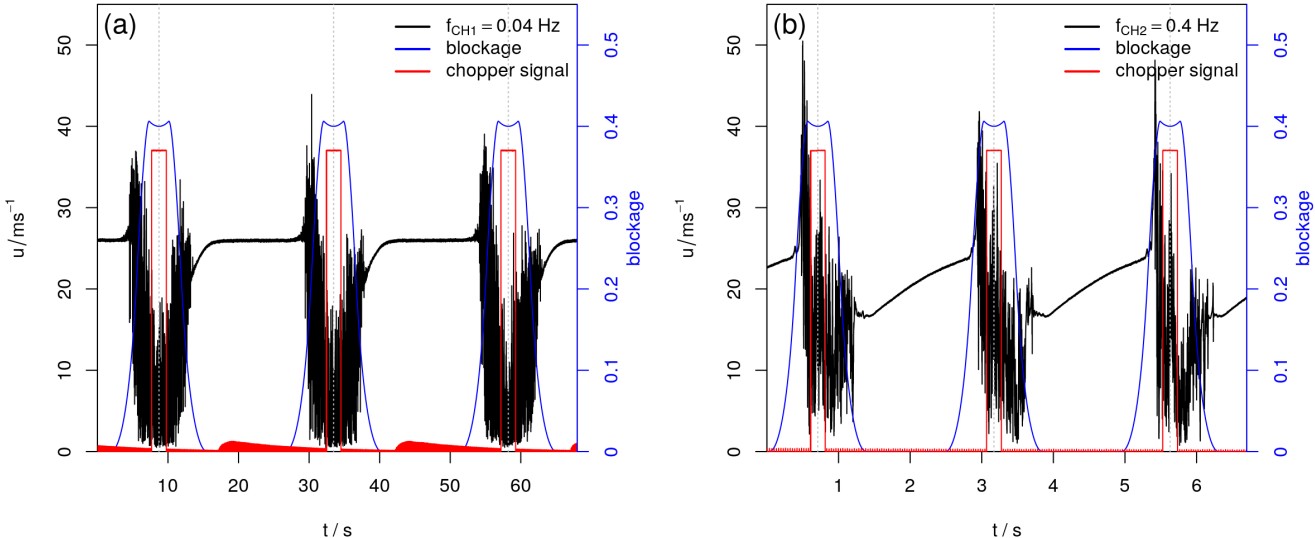

**Figure 3.** Gust produced by the chopper, measured at the centerline in $5.425\,w_{CH}$ distance from the chopper blade: (a) $f_{CH1} = 0.04\,\text{Hz}$, (b) $f_{CH2} = 0.4\,\text{Hz}$. The blue curve indicates the blockage, the red curve the signal from the induction sensor, and the gray dashed lines the moment when the chopper blade is horizontally aligned with the inlet.

### 3.1 Analysis of the time series

In figure 3, a part of the time series (black) that includes three inverse gusts is plotted for both chopper frequencies. In addition, the blockage induced by the chopper (blue) and the signal from the induction sensor (red) are plotted. The measurements were carried out at the centerline in $5.425\,w_{CH}$ distance from the chopper blade. Looking at figure 3(a) ($f_{CH1} = 0.04\,\text{Hz}$), it can be seen that the velocity first increases when the chopper blade enters the inlet. The velocity decreases then drastically and fast when the blockage of the inlet due to the chopper blade increases, and it recovers eventually after the chopper blade left the

inlet. Between the inverse gusts, the velocity equals the inflow velocity and has a turbulence intensity of $0.3\%$ as in the empty test section. While the inverse gusts evolve similarly, differences in the turbulent fluctuations are present. In case of $f_{CH2}$, see figure 3(b), the result looks similarly with the difference that the velocity can not recover between the inverse gusts. Instead, it increases until the chopper blade enters the inlet anew to create the next inverse gust. This indicates that the recovery time of the mean velocity to $u_0$ between the gusts needs to be taken into account in future experiments. One possibility to avoid the

gusts to interact with each other at high chopper frequencies would be to change the duty cycle, i.e. to use a chopper blade with only one arm. In addition, the fluctuations induced by the chopper are stronger in case of the high rotation frequency and can reach values above $50\,\text{ms}^{-1}$ which is twice the inflow velocity.

Figure 4 shows the energy spectral density $E(f)$ plotted over the frequency $f$ of the two above presented time series measured at the centerline $5.425\,w_{CH}$ downstream of the chopper blade. At low frequencies, the spectrum is approximately constant

but superimposed with the peak of the chopper frequency and its harmonics. From $f \approx 10\,\text{Hz}$, the energy spectral density

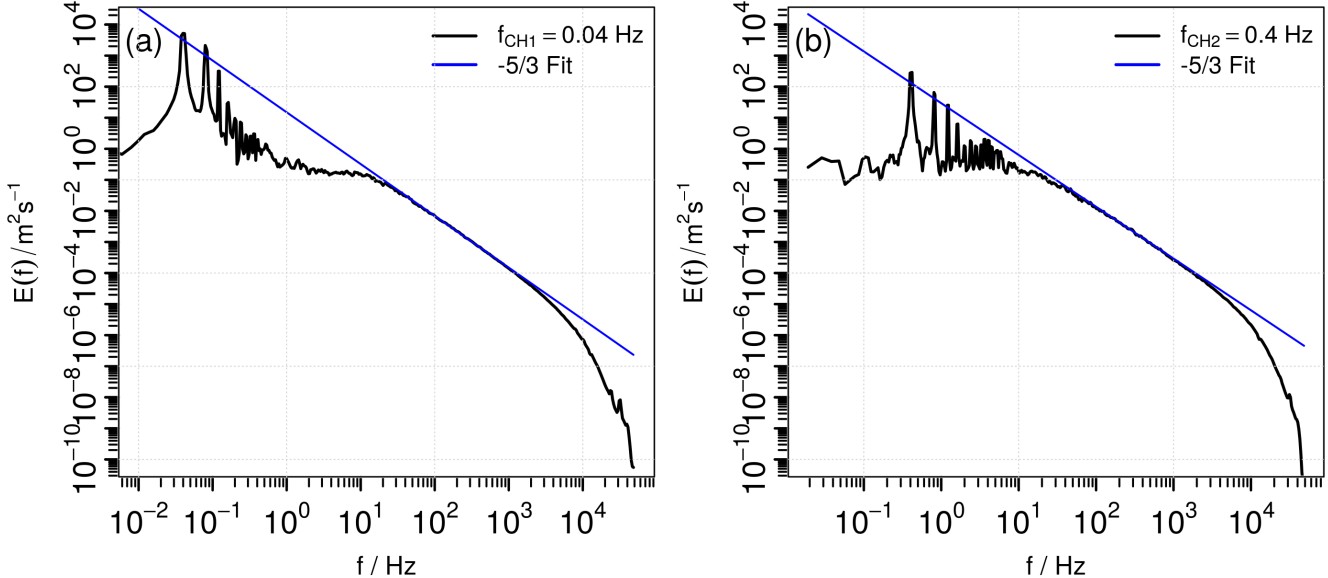

**Figure 4.** Energy spectral density of the measured time series at the centerline in $5.425\,w_{CH}$ distance from the inlet for the chopper frequencies (a) $f_{CH1} = 0.04\,$Hz and (b) $f_{CH2} = 0.4\,$Hz. The blue line indicates a power law decay according to $E(f) \propto f^{-5/3}$.

decays with increasing frequency according to $E(f) \propto f^{-5/3}$ up to $f \approx 1000\,$Hz. Kolmogorov predicted a scaling of the wave-number energy spectrum $E(k)$ according to $k^{-5/3}$ for "ideal" turbulence with dissipation in equilibrium (cf. Kolmogorov (1941a), Kolmogorov (1941b)). By means of Taylor's hypothesis of frozen turbulence, the wave-number spectrum can be converted to a spectrum in the frequency domain and the scaling becomes $f^{-5/3}$ (e.g. Laizet et al. (2015)). At first glance, it

is therefore tempting to interpret the scaling found in the presented energy spectra as an indication of the existence of features of homogeneous isotropic turbulence. However, as the measurements that are presented here are carried out in the highly turbulent near wake of the chopper blade, Kolmogorov's theory is not expected to hold. Other studies such as the one presented by Alves Portela et al. (2017) find a similar behavior of the turbulence. The origin of the $-5/3$ scaling may therefore be the intermittency of the turbulent flow, as for example Laizet et al. (2015) and Zhou et al. (2019) discuss.

The investigation of the gust time series and the energy spectral density shows how the recurring inverse gust generated by the chopper can be decomposed into three parts: the mean velocity of the inverse gust, the underlying shape of the inverse gust as recurring global large-scale flow structure, and the turbulence within the gust. This triple decomposition is illustrated in figure 5. The inverse gust (figure 5(a)) has a mean gust velocity $\bar{u}$ (figure 5(b)) with an underlying inverse gust shape $\tilde{u}$ (figure 5(c)) with strong turbulent fluctuations $u'$ (figure 5(d)), and $u_G = \bar{u} + \tilde{u} + u'$. The underlying inverse gust shape is here calculated by

phase-averaging the gust events and smoothing with a moving average with a window size of $0.1\,$s. The fluctuations are then extracted by subtracting the underlying inverse gust shape from the respective gust. As shown in figure 6, the energy spectrum of the whole time series $E_G(f)$, figure 6(a), can analogously be split into a high-energetic, low-frequent part $\tilde{E}(f)$ from the

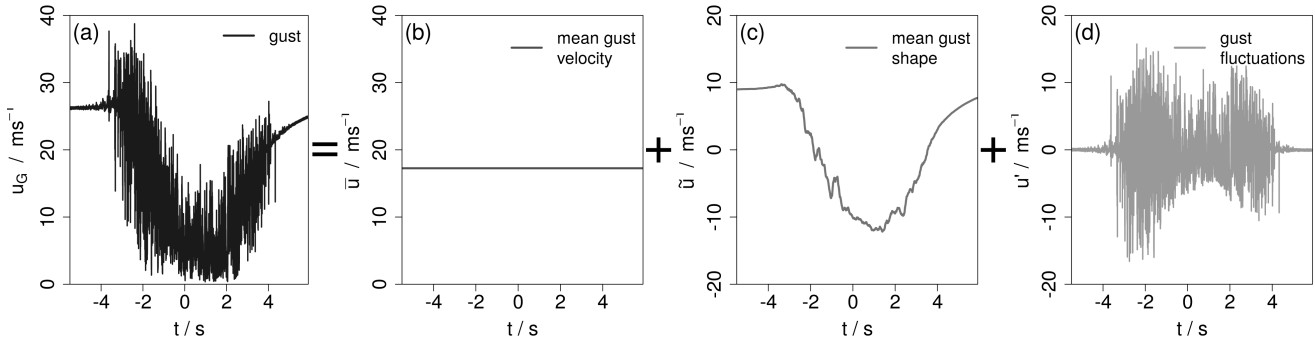

**Figure 5.** Example of a triple decomposition of the inverse gust $u_G = \bar{u} + \tilde{u} + u'$ (a) into its mean velocity $\bar{u}$ (b), the underlying inverse gust shape $\tilde{u}$ (c), and the fluctuations $u'$ (d).

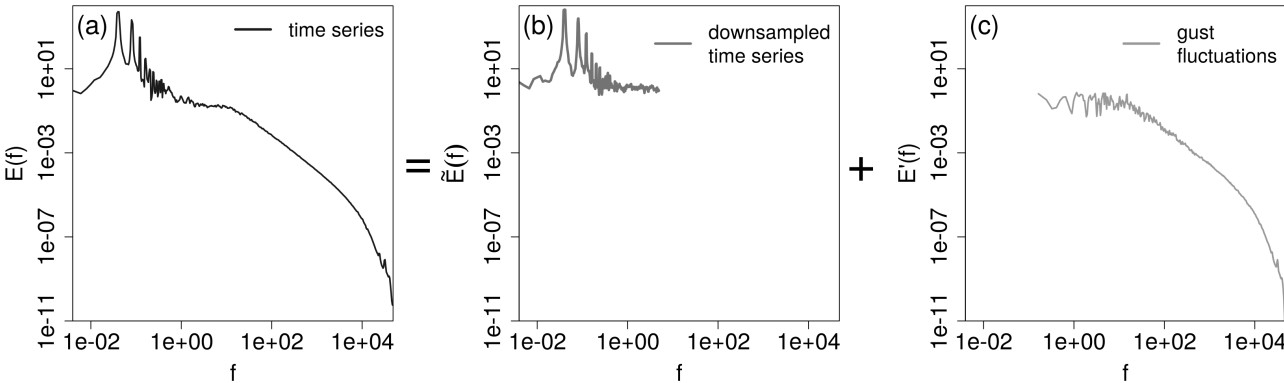

**Figure 6.** Example of a spectral decomposition of the inverse gust $E_G(f) = \tilde{E}(f) + E'(f)$ (a) into its periodic, high-energetic part $\tilde{E}(f)$ (b) and the spectrum of the fluctuations $E'(f)$ (c).

periodically recurring underlying inverse gust shape (figure 6(b)), and a high-frequent part from the gust fluctuations $E'(f)$ (figure 6(c)). To calculate $\tilde{E}(f)$, the time series was sampled down to $f'_s = 10\,\mathrm{Hz}$ by taking every 10000th data point. We use this method instead of applying a filter to emphasize the two distinguishable parts the spectrum is made of, and the frequency was chosen to have a small overlap between $\tilde{E}(f)$ and $E'(f)$. In case of $E'(f)$, the spectra of $u'$ of all inverse gust events within the time series are calculated and afterwards, the spectra are averaged.

### 3.2 Mean gust flow field: $\bar{u}$

In the following, the properties of the mean gust velocity flow field will be investigated by looking at the evolution of $\bar{u}$ across the measured planes. The mean gust velocity is calculated by averaging the individual inverse gust events within a time series and then taking the average over all values[3]. The average standard deviation is determined similarly. For a better comparison

---

[3]Note that the mean gust velocity differs by this definition from the mean velocity of the whole time series

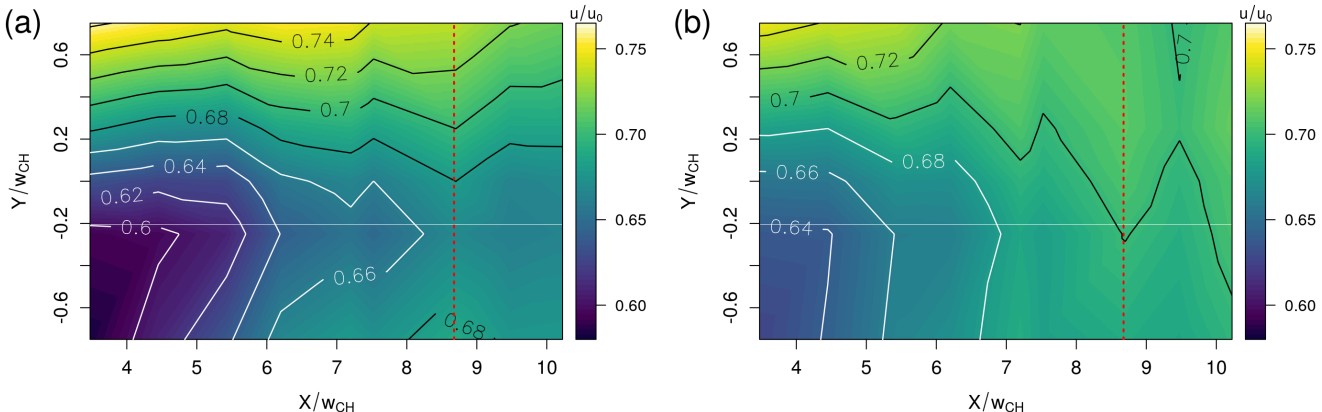

**Figure 7.** Interpolated surface plot of the downstream evolution of the mean velocity of the inverse gust for (a) $f_{CH1} = 0.04$ Hz and (b) $f_{CH2} = 0.4$ Hz. The red dashed line marks the downstream position of the YZ-plane.

between the measurements, the same window with respect to the chopper position is used for all data sets for the respective chopper frequency.

In figure 7, the downstream evolution of the average mean gust velocity that is normalized by the inflow velocity is presented as interpolated surface plot for both chopper frequencies. For both cases, the mean gust velocity evolves similarly downstream in the test section. Close to the inlet, the flow is inhomogeneous along the span-wise direction. The lowest velocities are found at the root side of the chopper blade at $X/w_{CH} = 3.475$ and $Y/w_{CH} = -0.75$. At the tip side of the chopper blade at $X/w_{CH} = 3.475$ and $Y/w_{CH} = 0.75$, the velocities are highest. A velocity difference of $0.14u_0$ (or $3.5$ ms$^{-1}$) in case of $f_{CH1} = 0.04$Hz and $0.08u_0$ (or $2.0$ms$^{-1}$) for $f_{CH2} = 0.4$Hz, respectively, is found in span-wise direction at $X/w_{CH} = 3.475$. With increasing downstream position, the flow field becomes more homogeneous in span-wise direction and the average gust velocity reaches approximately 70% of the inflow velocity at the end of the measurement plane. While the flow fields evolve in principle similarly, the velocities are higher and the flow field is less asymmetric in case of the high chopper frequency.

In figure 8, the mean gust velocity that is normalized by the inflow velocity is plotted in the YZ-plane at $X/w_{CH} = 8.675$ in a similar manner. In case of $f_{CH1}$, the velocity is lowest in the upper left corner ($Y/w_{CH} = -0.75$, $Z/w_{CH} = 0.75$) where the chopper blade enters the inlet, and highest in the bottom right corner ($Y/w_{CH} = 0.75$, $Z/w_{CH} = -0.75$) where the chopper blade leaves the inlet first. In case of $f_{CH2}$, the velocity is higher in the right half ($Y > 0$) than in the left half ($Y < 0$).

Next, the average turbulence intensity of the inverse gust is plotted. For this, the average over the standard deviations $\sigma_i$ of the $N$ inverse gusts within the time series is calculated, $\bar{\sigma}_G = \frac{1}{N}\sum_{i=1}^{N}\sigma_i$, and divided by the average gust velocity, $TI_G = \bar{\sigma}_G/\bar{u}$. Note that this gives a global turbulence intensity that takes the inverse gust shape and the fluctuations into account. The result is presented as interpolated contour plot in figure 9. In case of the lower chopper frequency, the highest turbulence intensities are found closest to the inlet, and the maximum turbulence intensity, $TI_{G,max1} = 60\%$, is measured at $X = 3.475\,w_{CH}$ and $Y/w_{CH} = -0.25$. The values are very high because the global turbulence intensity is used. The local turbulence intensity is in the order of magnitude of 25%. The flow field is asymmetric but becomes more homogeneous farther downstream when the

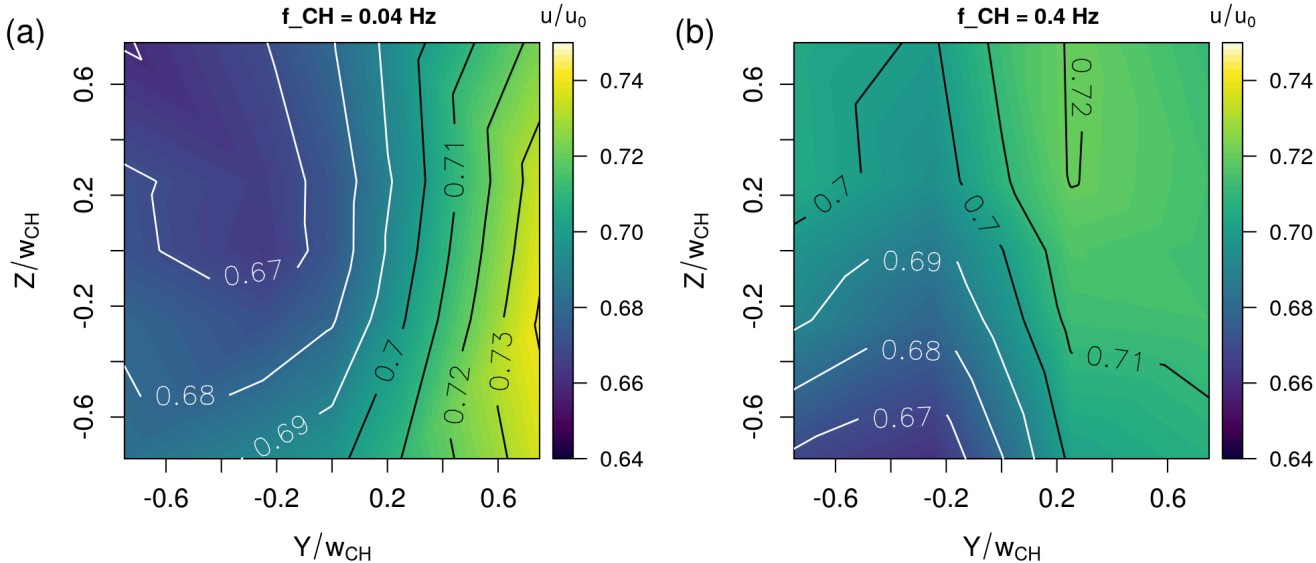

**Figure 8.** Interpolated surface plot of the mean velocity of the inverse gust in the YZ - plane $8.675 w_{CH}$ downstream of the chopper blade for (a) $f_{CH1} = 0.04\,\text{Hz}$ and (b) $f_{CH2} = 0.4\,\text{Hz}$.

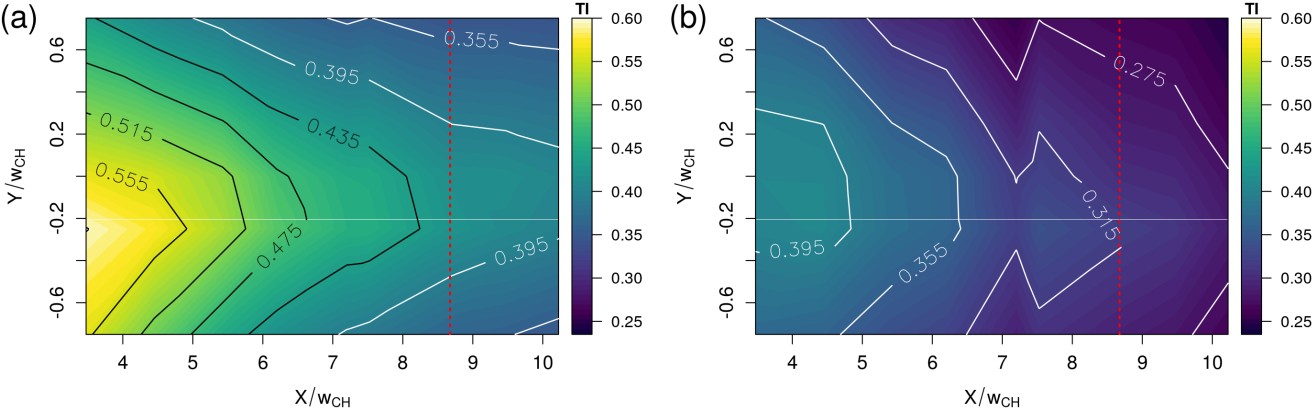

**Figure 9.** Interpolated surface plot of the turbulence intensity of the inverse gust for (a) $f_{CH1} = 0.04\,\text{Hz}$ and (b) $f_{CH2} = 0.4\,\text{Hz}$. The red dashed line marks the downstream position of the YZ-plane.

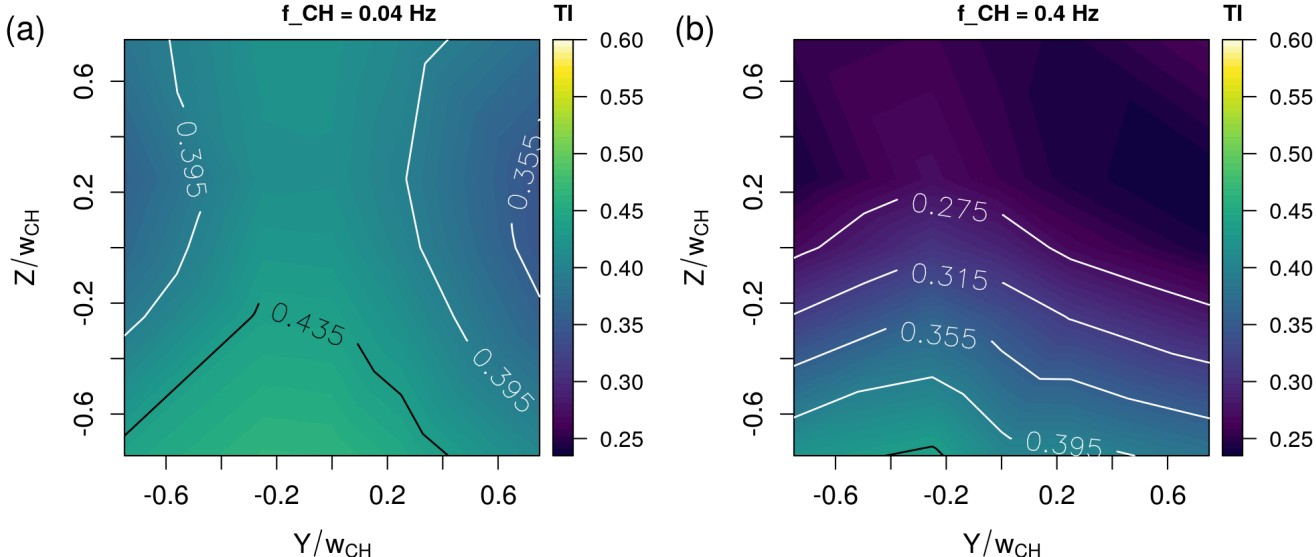

**Figure 10.** Interpolated surface plot of the turbulence intensity of the inverse gust in the YZ - plane $8.675w_{CH}$ downstream of the chopper blade for (a) $f_{CH1} = 0.04\,\text{Hz}$ and (b) $f_{CH2} = 0.4\,\text{Hz}$.

turbulence intensity decreases. The lowest value, $TI_{G,min1} = 34\%$, is found at $X/w_{CH} = 10.225$ and $Y/w_{CH} = 0.75$. While the evolution as such looks similarly in case of the high chopper frequency, the turbulence intensities are lower, decreasing from $TI_{G,max2} = 41\%$ at $X/w_{CH} = 4.45$ and $Y/w_{CH} = -0.25$ to $TI_{G,min2} = 24\%$ at $X/w_{CH} = 10.225$ and $Y/w_{CH} = 0.75$. As presented in figure 10, the average gust turbulence intensity does also vary spatially in the YZ-plane ($X/w_{CH} = 8.675$). In case of $f_{CH1}$ (figure 10(a)), the turbulence intensity is lowest at the vertical wind tunnel walls in the upper half of the measurement field and highest in the center in the bottom half. For the high chopper frequency, the average gust turbulence intensity is lowest in the top half of the measurement field and increases in the bottom half.

### 3.3 Underlying inverse gust shape: $\tilde{u}$

In the following, the underlying inverse gust shape $\tilde{u}$ will be investigated. For this, only the inverse gust segments of each time series are taken into consideration. Exemplarily, in figure 11, all inverse gusts from the respective time series measured at the two chopper frequencies and at the centerline at $X = 5.425\,w_{CH}$ are plotted normalized by the inflow velocity over the time $t$ that is normalized by the chopper period $T_{CH} = 1/f_{CH}$. Here $t/T_{CH} = 0$ denotes the horizontal alignment of the chopper blade in the inlet. In black, the smoothed average inverse gust is plotted[4], and in blue, the blockage of the wind tunnel is added. For both chopper frequencies, the single inverse gusts show strong fluctuations. The maximum velocity fluctuations can reach $1.8u_0$ (or $u \approx 45\,\text{ms}^{-1}$) ($f_{CH1} = 0.04\,\text{Hz}$) or $2.16u_0$ (or $u \approx 54\,\text{ms}^{-1}$) ($f_{CH2} = 0.4\,\text{Hz}$). While the inverse gust

---

[4]In case of $f_{CH1}$, a moving average with a window of $0.05\,\text{s}$ is applied, and in case of $f_{CH2}$, a moving average with a window of $0.005\,\text{s}$ is applied.

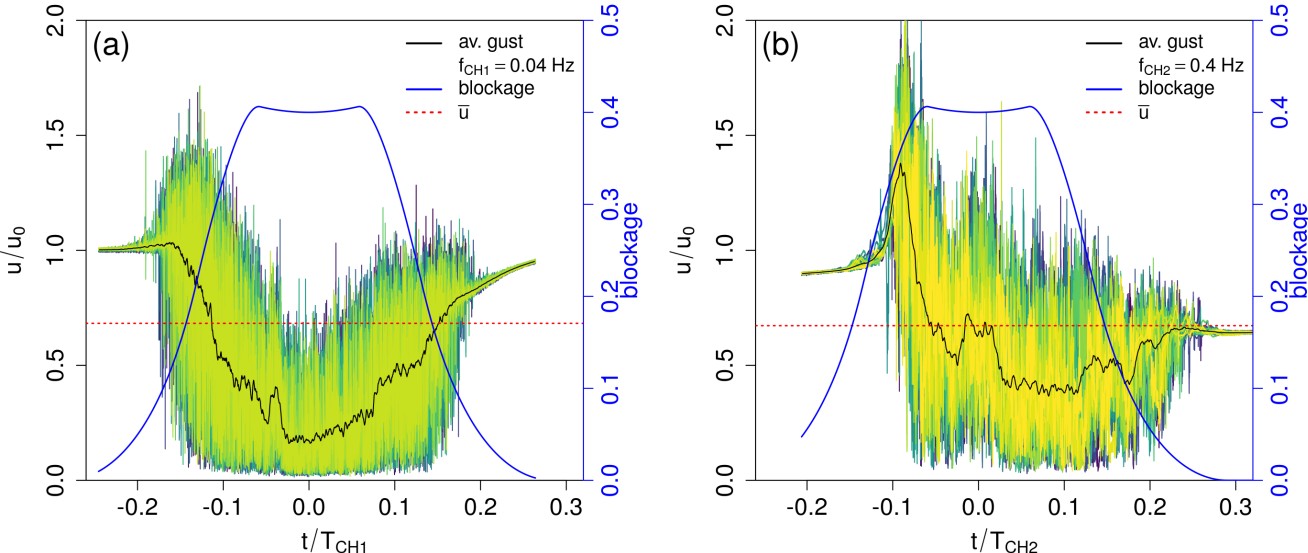

**Figure 11.** Plot of all inverse gusts captured in the time series measured at the centerline $5.425\,w_{CH}$ downstream the chopper blade for a chopper frequency of (a) $f_{CH1} = 0.04\,\text{Hz}$ and (b) $f_{CH2} = 0.4\,\text{Hz}$. In different colors (dark blue-green-yellow color map), the inverse gust events are plotted, in black, the smoothed average inverse gust is marked and in blue, the blockage is added with the corresponding axis on the right side.

generated with a chopper frequency of $f_{CH1} = 0.04\,\text{Hz}$ shows a smaller increase of the velocity in the beginning and is mainly characterized by a strong velocity deficit, in case of the faster chopper frequency, $f_{CH2} = 0.4\,\text{Hz}$, the increase in the beginning is much stronger and the velocity deficit afterwards less pronounced. The strong velocity increase can be explained by the faster blockage of the inlet in case of $f_{CH2}$: As the mass flow rate is expected to be similar, the faster blockage forces the flow to adapt quicker which leads to a sudden acceleration. Because the flow has more time to adapt in case of $f_{CH1}$, only little to no acceleration is present. For both chopper frequencies, the highly unsteady flow induces additionally strong fluctuations, and the combination of these strong fluctuations with the acceleration of the flow in case of $f_{CH2}$ leads to those extreme velocity peaks. In both cases, a small velocity increase around $t/T \approx 0.03$ is present. Its origin may lay in a flow change occurring when the chopper blade leaves the upper corner at the inner side of the inlet: The chopper blade rotates and progressively blocks the flow at the inlet of the test section. Once the blade has fully entered the inlet, the upper part of the test section is gradually unblocked, starting at the upper inner corner. The pressurized air is then abruptly released towards the low pressure area behind the chopper blade which leads to a sudden increase of the velocity as the flow is entrained into the wake of the blade before the flow adapts. Similar phenomena may happen when the chopper blade leaves the inlet, which is also marked by fluctuations of the velocity. As the flow has more time to adapt in case of $f_{CH1}$, the increase is less pronounced.

To increase the understanding of the underlying inverse gust shape $\tilde{u}$ across the width of the wind tunnel, figure 12 shows the smoothed average inverse gust that is normalized by the inflow velocity and plotted over $t/T_{CH}$ for all span-wise positions

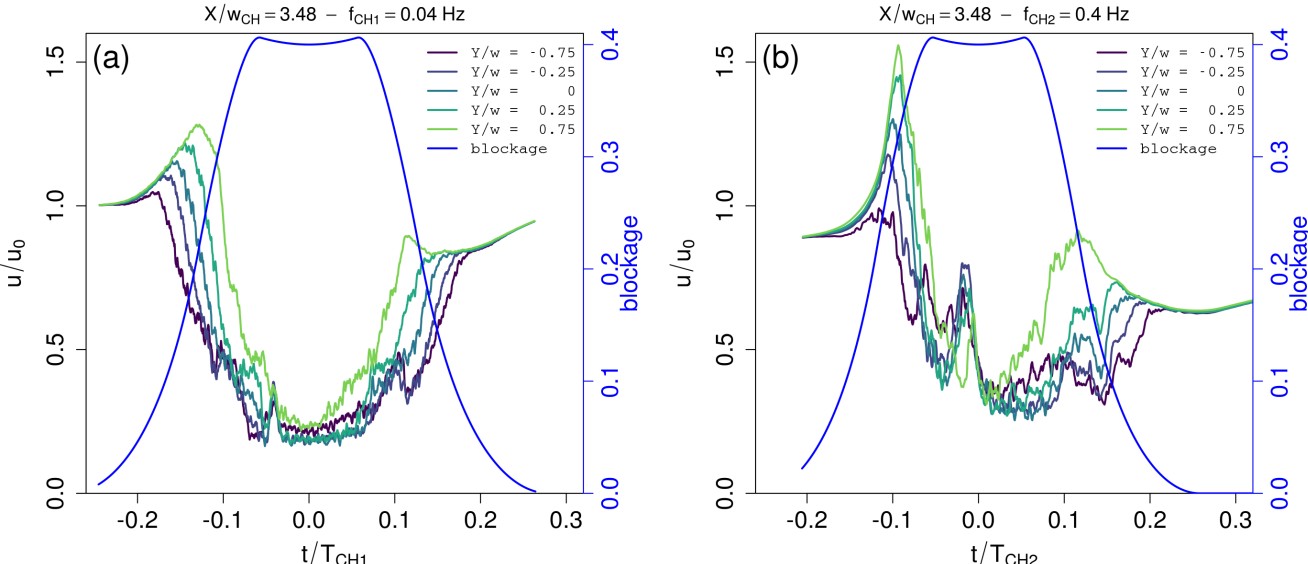

**Figure 12.** Span-wise evolution of the smoothened average inverse gust at $X/w_{CH} = 3.475$ with respect to the chopper blade position: The inverse gust as measured by the respective sensor is plotted over the normalized time where $t/T_{CH} = 0$ indicates that the chopper blade is aligned horizontally in the inlet, and the blockage is added in blue with the blue axis on the right side. The chopper frequencies are (a) $f_{CH1} = 0.04$ Hz and (b) $f_{CH2} = 0.4$ Hz.

and both chopper frequencies at $X/w_{CH} = 3.475$. In case of a low chopper frequency (figure 12(a)), the inverse gust shape itself is similar for all span-wise positions. The brief increase of the velocity that exceeds the inflow velocity is followed by a rapid velocity decrease ($18\,\mathrm{ms}^{-1}$ in 2.5 s which is significantly stronger than the decrease of $9.8\,\mathrm{ms}^{-1}$ in 2.8 s given as example for an EOG according to the IEC-61400-1 norm). Then, there is a slight and short velocity increase around $t/T \approx 0.03$,

and the velocity recovers once the chopper blade leaves the inlet. From $Y/w_{CH} = -0.75$ where the chopper blade enters the test section first with the the blade root side, to $Y/w_{CH} = 0.75$ where the chopper blade enters the test section last with the blade tip side, the characteristic time of the inverse gust decreases and its amplitude increases. This is in agreement with the interpretation that a faster blockage leads to a stronger increase of the velocity at the beginning ot the inverse gust: The change of the blockage is faster at the tip side of the chopper blade than at the root side due to the radially changing velocity of the

chopper, and therefore, the amplitude of the inverse gust increases towards $Y/w_{CH} = 0.75$. In addition, the local blockage is at first higher at the inner side of the test section. This combination leads to a higher, later velocity peak at the outer side of the test section. The above-mentioned small, short increase in velocity around $t/T \approx 0.03$ is more pronounced at the measurement positions towards the inner side of the inlet. This is in agreement with the above-mentioned release of the pressurized air when the chopper blade gradually unblocks the upper side of the inlet while entering.

These observations also hold for the high chopper frequency (figure 12(b)) but in a more pronounced manner.

Moving downstream to the last measurement position at $X/w_{CH} = 10.225$ shows how the inverse gust is transported in stream-

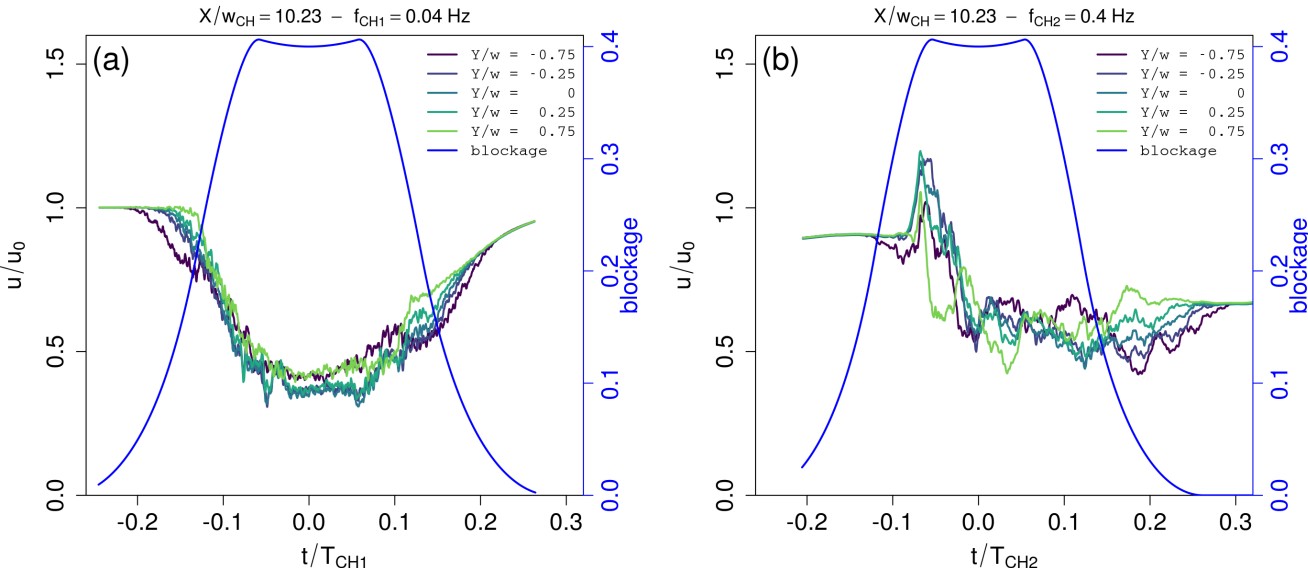

**Figure 13.** Span-wise evolution of the smoothed average inverse gust at $X/w_{CH} = 10.225$ with respect to the chopper blade position: The inverse gust as measured by the respective sensor is plotted over the normalized time where $t/T_{CH} = 0$ indicates that the chopper blade is aligned horizontally in the inlet, and the blockage is added in blue with the blue axis on the right side. The chopper frequencies are (a) $f_{CH1} = 0.04\,\mathrm{Hz}$ and (b) $f_{CH2} = 0.4\,\mathrm{Hz}$.

wise direction (cf. figure 13). In case of the low chopper frequency (figure 13(a)), the inverse gust disperses, i.e. the characteristic time increases while the amplitude decreases. The minimal velocity within the inverse gust increases while the velocity peak is not present anymore. Additionally, the small, quick velocity increase at $t/T \approx 0.03$ can not be distinguished anymore.

Also, the underlying inverse gust shape homogenizes and evolves similarly for all span-wise positions. In comparison, the underlying inverse gust shape imprinted onto the flow with a higher chopper frequency (cf. figure 13(b)) preserves the variation in span-wise direction. While the structure disperses as well, the velocity peak at the beginning of the structure is still visible. Due to the downstream transport of the inverse gust, the velocity peaks are found later with respect to the radial chopper blade position that is indicated by the blockage.

With a chopper blade width of $w_{CH} = 20\,\mathrm{cm}$ as presented in this setup, the inverse gust amplitude is higher than proposed by the IEC-64100-1 norm. The changes in the velocity can be rapid, for example approximately $18\,\mathrm{ms}^{-1}$ in $2.5\,\mathrm{s}$ in case of $f_{CH1} = 0.04\,\mathrm{Hz}$ and up to $25\,\mathrm{ms}^{-1}$ in $0.25\,\mathrm{s}$ in case of $f_{CH2} = 0.4\,\mathrm{Hz}$.

In order to quantify the characteristic time of the inverse gust and its dispersion, figure 14 shows the downstream evolution of the gust characteristic time $\Delta t/T_{CH}$ that is normalized by the chopper period for both chopper frequencies and all hori-

280 zontal measurement positions. The gust characteristic time is calculated by taking the absolute value of the normalized gust fluctuations $u'/\bar{u}$ to find the interval in which the fluctuations frequently exceed a threshold set to be 0.025. This threshold was chosen to be low enough to capture the beginning of the gust well while also being high enough to neglect possible

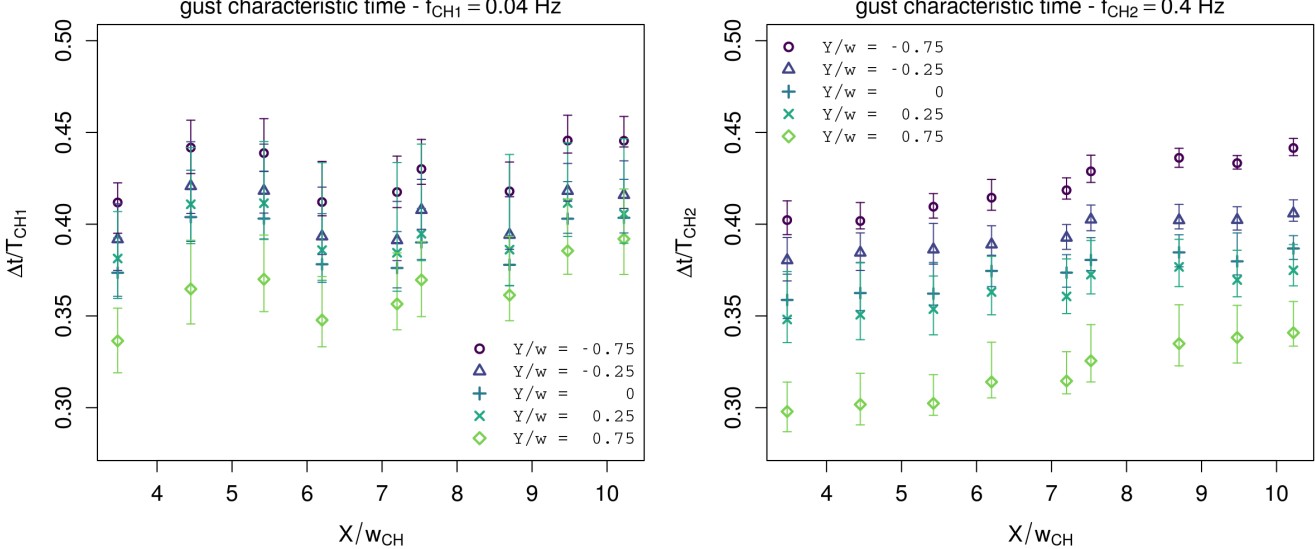

**Figure 14.** Average normalized characteristic time $\Delta t / T_{CH}$ of the gusts at the respective span-wise and downstream positions for (a) $f_{CH1} = 0.04\,\mathrm{Hz}$ and (b) $f_{CH2} = 0.4\,\mathrm{Hz}$.

unexpected dynamics in the laminar part between the gust. In addition, the thresholds 0.02 and 0.03 were used to determine errors from the sensitivity of this method towards different thresholds. The results are plotted in form of error bars that may be asymmetric. For both chopper frequencies, the gust characteristic time increases with increasing distance from the inlet as the gust disperses downstream. Additionally, the gust characteristic time is longer for the inner span-wise measurement positions $Y/w_{CH} = -0.75$ and $Y/w_{CH} = -0.25$ than for the outer span-wise positions ($Y \geq 0$), because the root side of the chopper blade passes by slower than the tip side. However, with respect to the errors, the gust characteristic times are similar in case of $f_{CH1}$. Quantitatively, the gust characteristic time scales anti-proportionally with the chopper frequency, which is nicely shown by the similar normalized gust characteristic times for the two chopper frequencies. The approximate gust characteristic times are $\Delta t(f_{CH1}) \approx 10\mathrm{s}$ and $\Delta t(f_{CH2}) \approx 0.9\mathrm{s}$. $\Delta t(f_{CH1}) \approx 10\mathrm{s}$ matches the characteristic time of the IEC EOG $\Delta t_{IEC} = 10.5\mathrm{s}$ closely. $\Delta t(f_{CH2}) \approx 0.9\,\mathrm{s}$ is scaled to wind tunnel dimensions: The ratio of chord length $c$ and gust characteristic time should be constant which is achieved for the airfoil that we will use with $c = 0.09\,\mathrm{m}$.

### 3.4 Gust fluctuations: $u'$

After investigating the mean gust properties and the underlying inverse gust shape, next, some turbulence characteristics of the gust will be discussed by investigating the gust fluctuations $u'$. Figure 15 shows the energy spectral density of the turbulent fluctuations $u'$ (cf. figure 11 (d)) of all individual inverse gusts captured in the above-discussed time series measured at $X/w_{CH} = 5.425$ for the respective chopper frequencies. In addition, the average over all spectra is plotted in red, and the

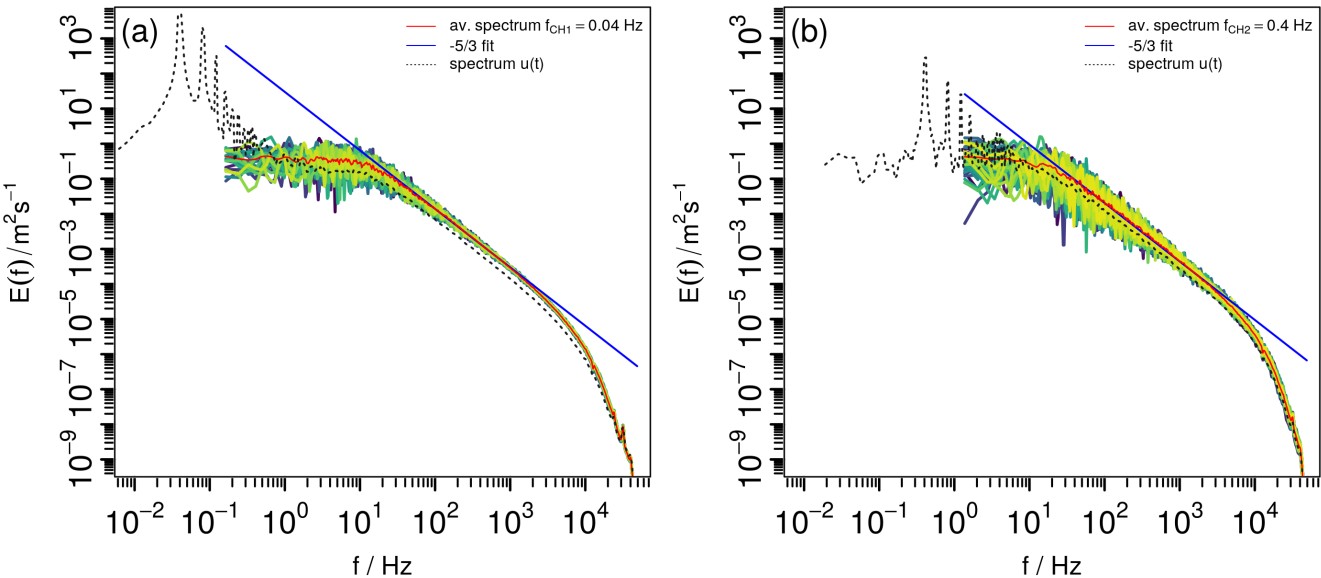

**Figure 15.** Energy spectral density of all inverse gusts (blue-green-yellow color map) with the average spectrum plotted in red and a decay according to $E(f) \propto f^{-5/3}$ plotted in blue. The downstream position is $X = 5.425\, w_{CH}$, the measurement was carried out at the centerline and the chopper frequency is (a) $f_{CH1} = 0.04\,\mathrm{Hz}$ and (b) $f_{CH2} = 0.4\,\mathrm{Hz}$.

decay according to $E(f) \propto f^{-5/3}$ in the inertial sub-range is indicated in blue. The plots show that the turbulence within the inverse gust has approximately the same energy spectrum for all inverse gusts. The inertial sub-range follows a decay according to $E(f) \propto f^{-5/3}$ in the frequency range between $20\,\mathrm{Hz} \leq f \leq 2000\,\mathrm{Hz}$ ($f_{CH1} = 0.04\,\mathrm{Hz}$) and $20\,\mathrm{Hz} \leq f \leq 5000\,\mathrm{Hz}$ ($f_{CH2} = 0.4\,\mathrm{Hz}$), respectively. As above-mentioned, we do not expect Kolmogorov's theory to hold in this flow which leads to the interpretation that the origin of the -5/3 scaling of the inertial sub-range may be the intermittency of the flow (Laizet et al. (2015),Zhou et al. (2019)).

To further characterize the gust turbulence, the one-dimensional energy spectrum is as suggested by Hinze (1975) used to calculate the integral length scale $L = \lim\limits_{f \to \infty} \left( \frac{E(f) \cdot \overline{u}}{4\sigma^2} \right)$ of the turbulence. The integral length scale presents here the large scales within the turbulence which supplements the gust characteristic time as global scale. The results are shown for both chopper frequencies as interpolated contour plots in figure 16. While in case of the lower chopper frequency $f_{CH1} = 0.04\,\mathrm{Hz}$, higher values of $L$ are present, the evolution of $L$ across the measurement plane behaves in principle similarly: $L$ increases both in span-wise direction from $Y/w_{CH} = -0.75$ to $Y/w_{CH} = 0.75$ and downstream from $X/w_{CH} = 3.475$ to $X/w_{CH} = 10.225$. With respect to the average integral length $\bar{L}$ over the whole flow field, the gradient in span-wise direction is approximately 20% in case of $f_{CH1}$ and between 20% ($X/w_{CH} = 3.475$) and 0.6% ($X/w_{CH} = 10.225$) in case of $f_{CH2}$. The downstream gradients are 63% and 40%, respectively, which indicates that the change in stream-wise direction is more significant. The smallest values are found at $X/w_{CH} = 3.475$ and $Y/w_{CH} = -0.75$ with $L = 0.25\, w_{CH}$ in case of $f_{CH1} = 0.04\,\mathrm{Hz}$ and $L = 0.245\, w_{CH}$

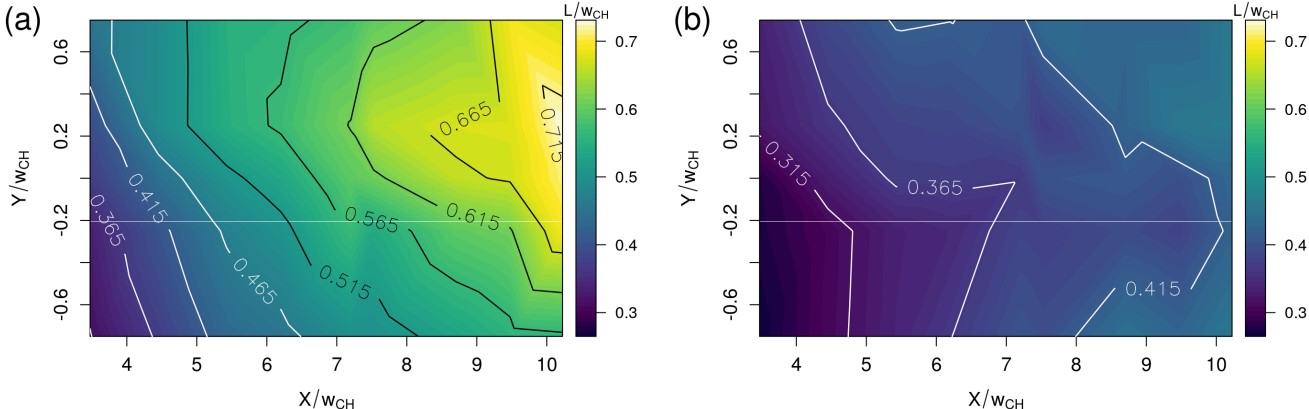

**Figure 16.** Interpolated surface plot of the integral length scale of the gust fluctuations for (a) $f_{CH1} = 0.04\,\text{Hz}$ and (b) $f_{CH2} = 0.4\,\text{Hz}$.

in case of $f_{CH2} = 0.4\,\text{Hz}$, respectively. As the turbulence evolves downstream, $L$ increases and the highest values are found at the opposite corner of the flow field at $X/w_{CH} = 10.225$ and $Y/w_{CH} = 0.25$ with $L = 0.73\,w_{CH}$ in case of $f_{CH1} = 0.04\,\text{Hz}$ and $L = 0.46\,w_{CH}$ in case of $f_{CH2} = 0.4\,\text{Hz}$.

The integral length scale is of interest when carrying out aerodynamic experiments because the interaction between the airfoil
and the flow structures depends on the ratio $L/c$. It is for example indicated in Cao et al. (2011) that a smaller integral length in the flow delays stall.

## 4   Conclusions and Outlook

A new system for the generation of strong, sudden inverse gusts in a wind tunnel, the chopper, has been presented together with a first investigation of the flow generated by it. The experimental campaign was performed at 65 measurement points in
the three-dimensional volume of the test section to characterize the stream-wise flow behind this new unsteady perturbation system. The focus of this study was to find governing parameters that influence the flow to refine experiments in the future. The inverse gust structure induced by the chopper has been examined for a fixed inflow velocity and two different chopper frequencies, and the parameters were chosen to match the specifications of future experiments. First, the periodic flow variation that is generated each time the chopper blade crosses the inlet was discussed by means of the time series and its energy spectrum.
To gain a better understanding of the inverse gust structure itself, in the next step, the gust events were brought into focus. It was shown how the inverse gust has an underlying shape with superimposed fluctuations, so that each inverse gust can be decomposed into a mean gust velocity $\bar{u}$, the global underlying inverse gust shape $\tilde{u}$ and the high-frequent fluctuations $u'$. The decomposition can analogously be carried out in the spectral domain. For the further analysis, this triple decomposition was used to discuss the downstream evolution of the underlying inverse gust shape that disperses downstream while the average
gust velocity increases and the global turbulence intensity decreases downstream. The average flow field is asymmetric close to the chopper blade but becomes more homogeneous farther downstream. Some aspects of the underlying inverse gust shape

were related to the physical behavior of the system, and it was explained that some mechanisms responsible for the inverse gust generation appear to be universal with respect to the chopper frequency. The gust characteristic time is anti-proportional to the chopper frequency and the investigated chopper frequencies generate inverse gusts with characteristic times similar to $\Delta t_{IEC}$ and downscaled to wind tunnel dimensions where a constant ratio of airfoil chord length to gust characteristic time is aimed for. The mean gust amplitude is larger than the one proposed in the IEC-64100-1 for all measurements. In addition, the velocity can change rapidly. The fluctuations were analyzed by means of the energy spectrum that shows a turbulence decay with an inertial sub-range that decays according to $E(f) \propto f^{-5/3}$, and an integral length scale that increases downstream as the turbulence evolves and that is in the order of magnitude of half the chopper blade width. The inverse gust can therefore be characterized by a global scale, namely its characteristic time, and an outer turbulence scale, namely the integral length.

Overall, a new system was presented that is capable of generating large, rapid velocity fluctuations while also inducing turbulence. By means of the downstream position and the chopper frequency, experiments can be executed in different regions of the flow field with different characteristics and different complexity that increases towards the inlet. For example, the aerodynamic behavior of an airfoil exposed to the flow can be studied downstream where the flow is already quite homogeneous, and then, it can be compared to the aerodynamic behavior of the airfoil positioned farther upstream in the inhomogeneous region where three-dimensional effects will be present. This gives an opportunity to study the effect of radial inflow changes which may cause similar effects as rotational augmentation that is observed in rotational blades due to an additional radial flow (cf. Bangga (2018)).

This investigation has shown us limits and opportunities of the setup and the measurements which helps us to improve the setup in the future. The currently very high blockage induced by the chopper blade will be reduced in the future by reducing the chopper blade width. We expect this to also decrease the currently very high gust amplitudes. Also, new chopper blade designs are possible that reduce the inhomogeneity of the flow field. To generate "normal" instead of inverse gusts, a design of a rotating disc with varying blockage is also possible. As the flow between the gusts has a low turbulence intensity, in addition, background turbulence can be added by installing a grid upstream of the chopper blade. To verify to what extent the gust generation mechanisms are universal with respect to the chopper frequency, more chopper frequencies need to be investigated. The experimental campaign that was presented gives a good overview of the flow evolution in the test section, but it should be complemented by measurements of all flow components and with higher spatial resolution of the measurement points. To form a comprehensive picture both in space and in time, future investigations may involve phase-averaged and/or time resolved particle image velocimetry as well as simulations.

*Author contributions.* System design, C. B.; Data acquistion, I. N.; Formal analysis, I. N.; Funding acquisition, C. B.; Investigation, I. N.; Methodology, I. N.; Project administration, C. B.; Resources, C. B.; Writing - original draft, I. N.; Writing - review & editing, I.N. and C. B.

*Competing interests.* The authors declare no conflict of interest

*Acknowledgements.* This project is funded by WEAMec, Pays de la Loire, Ecole Centrale Nantes and Nantes Metropole. Further, the authors would like to thank CSTB for providing measurement equipment.

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
