# Peer review of "First characterization of a new perturbation system for gust generation: The Chopper"

_Wind Energy Science, 2019_

## Referee Comment (RC1) · Anonymous Referee #1 · 4 Feb 2020

**General Comments:**

In this work, the authors introduce and characterize a new method for generating large gust disturbances in a wind tunnel using a rotating blade. Two gust frequencies were tested, and hot-wire measurements showed large gust amplitudes in both cases. The gust signals were decomposed into mean, disturbance, and turbulence components, and each of these components was analyzed in detail. Overall, the study is well motivated, designed, and documented, and the capability of the "chopper" to generate significant, repeatable gust events is conveyed convincingly. I would prefer, however, that the aerodynamics of the device be considered in more detail. Since only two gust frequencies are studied, a more comprehensive explanation of the physical principles that govern the gust generator's influence on the flow would greatly strengthen the

characterization in this paper. Thus, while the gust-generation mechanism itself is novel and of interest to the wind-energy community, I believe a more detailed analysis of the results is required before this paper can be accepted for publication.

**Specific Comments:**

S1: [Section 1] The introduction motivates the problem well, and the review of relevant literature is concise and effective. I note, however, that the EOG profiles in Figure 1 are never compared with the gust profiles induced by the "chopper" mechanism. The authors could make the connection between these classes of gust profiles clearer. It should also be specified that the gusts of interest to this study are in the streamwise direction.

S2: [Section 2] More information about the closed-loop wind tunnel should be given. Since the test section of the wind tunnel is presumably discontinuous due to the construction of the gust generator, statistics regarding the baseline flow conditions in the measurement area would help isolate the influence of the gust generator from any background effects (especially in light of the turbulence intensities presented later in the work, e.g. Page 5, Line 94). The limitation of this gust-generation mechanism to wind tunnels with discontinuous test sections should also be stated, or modifications to the design for fully closed test sections should be proposed.

S3: [Section 2] The design of the "chopper" system should be further motivated by aerodynamical considerations. Is the primary purpose of the mechanism to produce large fluctuations in blockage, in order to create large oscillations in the free-stream velocity? If so, is there a qualitative difference in the method to closing and opening an active grid? Is the rotational motion of the blade intended to induce additional flow-normal velocity fluctuations due to the shearing action of the blade on the incoming flow? If, as the authors point out in Figures 6 through 9, the resulting flows are asymmetric in the test section, why was a rotating system used instead of a rising and falling guillotine-like blade? The authors should explain why this particular design was chosen in light of their goal of creating large-amplitude turbulent gusts.

[Figure]

S4: [Section 2, Page 4, Line 78] Why were 20 gust events chosen for these experiments? An analysis of statistical convergence would make the study more comprehensive.

S5: [Section 3.1] Why is it that the higher rotation frequency leads to stronger gusts (Page 5, Lines 97-98)? Why do the gust velocities in this case exceed twice the inflow velocity? I highlight these observations as examples: the authors are very comprehensive in explaining their findings, but a physically motivated discussion of the reported phenomena is generally lacking. I would prefer that these observations be explained (and further analyzed as necessary), so that the generalizability of these findings can be inferred.

S6: [Section 3.1, Page 5, Line 101] Please comment on why an isotropic-turbulence assumption applies to this flow. Given the character of the forcing, I would expect that the generated turbulence would be significantly anisotropic.

S7: [Section 3.1] The decomposition of the gust into three components is helpful for the analysis of the results. Please specify the method by which the gust shape and turbulent fluctuations were isolated from the mean flow, as the method by which these were obtained affects the spectral signatures of the gust and fluctuation profiles. In Section 4, Page 14, Line 14, the authors apply a second-order filter to obtain the corresponding spectra. If this was the method used to achieve the initial decomposition, how was the cutoff frequency selected? Why wasn't a filter with a sharper roll-off used? These details are needed to explain whether the fluctuations in the gust profiles shown in Figures 11 and 12 are part of the mean gust profile or turbulent fluctuations that passed through the filter due to the relatively gentle roll-off of the second-order filter.

S8: [Section 3.2] The authors do a good job characterizing the mean flow fields downstream of the gust generator. Could the authors comment on the extent to which the spanwise flow asymmetry shown in Figures 6 through 9 would affect wind-tunnel experiments (e.g. of nominally 2D wind-turbine blade sections)?

S9: [Section 3.3] Another instance of my request in S5, above. Please explain the

qualitative trends described in the text. For example, why does the lower-frequency disturbance have a smaller increase in velocity at the beginning of the gust event than the higher-frequency disturbance (Page 9, Lines 149-150)?

S10: [Section 3.3] In Figure 11b, there is a strong spike in the gust profiles just before $t/T = 0$ that is not present in the corresponding profiles for the low-frequency case. Is this significant, or is it just an artifact of filtering (see S7, above)? If it is significant, what caused it? Its presence seems to suggest that different physics are involved in the high-frequency case. These dynamics should be discussed if the effects of the gust generator are to be inferred outside of the two frequencies tested in the experiments.

S11: [Section 3.3] In Figure 13a, the data points between $X/w_{CH} = 6$ and $9$ appear to be offset uniformly downward from the trend of the rest of the data. Is this significant? Error bars for these data points would be helpful in answering this question, and in demonstrating the significance of the trends.

S12: [Section 4] The authors explain that the gusts have 'outer' and 'inner' scales, based respectively on the global gust disturbance and the integral length scale of the generated turbulence. It would be helpful to compare their relative magnitudes in physical space. Is there a frequency at which the outer scale would match the inner scale, and if so, would the character of the gusts qualitatively change at this frequency?

S13: [Section 4, Page 14, Line 208] Could the offset in the spectra be attributed in some way to the filtering method?

**Technical Corrections:**
T1: The use of the backslash to denote units in plots could be confusing. Parentheses or brackets would be preferable.

T2: Minor corrections to the references: 1) The two Wei et al. (2019) references on Page 2, Line 31 are reversed. 2) The active-grid reference on Page 2, Line 38 should use the surname Makita, rather than the given name Hideharu. 3) The Wester et al.

(2018) and Petrovic et al. (2019) citations require conference information.

T3: The legend in Figure 5b may be clearer as "mean streamwise velocity", since "gust velocity" carries the connotation of a disturbance rather than a mean quantity.

T4: The contours and labels in Figures 8b, 9b, and 15b are hard to read. Can they be shown in white instead?

T5: The caption to Figure 10 is written somewhat confusingly. It should read "The gust events are plotted in different colors, the smoothed average gust is marked in black, . . .". The color map of the gust events should also be specified, as it is in the caption to Figure 14.

T6: There are a number of minor typographical and grammatical errors in the manuscript. They do not impact the legibility of the work (on the whole, it is very well written and logically constructed), but they should be addressed at some point.

---

## Referee Comment (RC2) · Anonymous Referee #2 · 4 Feb 2020

General:
A novel approach for generating high velocity fluctuations with a rotating chopper disc in a wind tunnel is presented. The effect of the disc is investigated for two different rotational velocities of the chopper at a constant wind speed. The absolute flow is measured detailed by a 1D hot wire array, which is traversed downstream. From this data the absolute velocity, the turbulence intensity and spectra are calculated. Furthermore, the author shows that each gust can be divided in the mean velocity, the gust shape and additional turbulence.

Overall, the study shows a very interesting approach for a new device to generate velocity fluctuations under controllable conditions. The study has been carried out very conscientiously and to a large extent. In my opinion, however, the results show that the

flow of the chopper needs to be investigated in more detail. The author shows just two different cases, which are compared to each other. Furthermore, the high turbulence intensities of up to 60%, as well as an absolute velocity minimum close to 0 m/s show that a hot wire is not a suitable measuring instrument for measurements near the chopper. The study would therefore benefit greatly from a measurement technique that can distinguish between positive and negative velocities. LDA measurements, for example, are suitable for this purpose. Also the used wind tunnel seems to have a huge problem with the high blockage induced by the chopper, which is indicated by the large recovery time of the velocity.

The approach of flow modulation upstream presented here is very interesting and the results presented here show a nice first insight, but in my opinion the results must be looked at more closely and above all more critically due to the used hot wires and the limited cases.

Specific Comments:

1. Section 1: The goal of the study is well motivated, as it is a current problem of the wind energy community. Also the current literature is well reviewed. However, the work of D. Greenblatt (e.g. Unsteady Low-Speed Wind Tunnels (2016)) should also be included in the literature, as here speed variations due blockage using a louver have already been worked with.

2. Section 1: The authors show in figure 1 the EOG which serves as motivation. Also, a duration of 10.5s is mentioned. Unfortunately, the authors do not compare their results of the chopper against those defined gusts. I am also missing the mentioned frequency of the gust in the presented results, or an explanation why this frequency was not used.

3. Section 2: The author should comment on the use of 1D hot wires since the wake of a flat plate is a very complex and three-dimensional flow.

4. Section 2 (P4 L76): The authors do not comment why the flow velocity of 25 m/s is used and why they choose the presented frequencies. They also choose to measure "more than 20 gust events" and do not comment why they choose to do so. In addition for the 0.4 Hz case the number of measured gusts is doubled compared to the 0.04Hz case. Were all data used for the evaluation?

5. Section 3 (P5 Fig 3): The data show a large recovery time of the flow velocity after the chopper leaves the measurement section. This is not discussed by the authors but shows that the wind tunnel is probably not suited for such high blockages.

6. Section 3.1 (P5 L97): For the case with higher rotation higher fluctuations are measured. The duplication of velocity do not match with the induced blockage. This indicates strong aerodynamic effects of the chopper since the measured spikes occur when the chopper enters the test section. I think the authors need to be careful with interpreting these peaks without knowledge of the ongoing effects.

7. Section 3.1 (P6 L103): The decomposition of the gust into mean velocity, fluctuations and turbulence is a nice approach. It should be complained in more detail, how the decomposition was calculated.

8. Section 3.2 (Fig 6  7): The flow field has been studied very thoroughly and the results are well presented. The results show nevertheless a high asymmetric shape. This seems to be correlated with the shape of the chopper and with the time the blade stays in the inlet of the wind tunnel. Are there plans to improve this with a different geometric chopper shape? How should one deal with such high asymmetries during experiments with model wind turbines or blades?

9. Section 3.3 (P10 Fig 10): The shown fluctuations seem to be cut at low velocities. Here it needs to be verified if the results are correct or if this is a result of the used

1D hot wire and the used calibration.

10. Section 3.3 (P10 Fig 10): The average gusts show for both cases a peak at t/T = -0.05. The author does not comment on this peak, but it seems to be present in all gusts, since it shows up in the averaged data. The peak is also present for different positions as it can be observed in Fig. 11 as well.

11. Section 3.3 (P11 Fig 13): It is explained how the gust duration is determined; however, I am missing error bars in the plot. Also an explanation why the threshold of 0.025 is used is not explained further, but is crucial for the calculated gust duration.

12. Section 3.4 : The integral length scales are calculated from the data, but is the information one could draw from this? I think if those numbers are presented the should be put in some context.

---

## Referee Comment (RC3) · Anonymous Referee #3 · 7 Feb 2020

This work describes the construction and output of a, to my knowledge, novel turbulence generator in a wind tunnel. This device is called "the chopper" and is essentially a spinning blade that cuts through, or "chops", the wind tunnel inlet. The blade can be spun at different frequencies, resulting in different intermittently turbulent flows in the wind tunnel test-section. One-component hot-wire measurements were performed in order to assess the chopper's output. It is demonstrated that the chopper imparts sudden changes on the centerline mean velocity, which is particularly useful as most other turbulence generating techniques cannot create a flow which responds so rapidly. Homogeneity and spectra are also investigated, demonstrating that the flow becomes more homogeneous as it develops downstream and that the rotational rate of the chopper blade and its harmonics are present at the low frequencies in the spectra, with a

turbulent looking spectrum at higher frequencies. Both of these observations are perhaps obvious, but nonetheless are shown for the first time here for this novel setup. One feature of the paper I particularly liked was that it often made me ask a question in my head, and I found that the answer was present later in the paper.

I think after some review, this work will be worthy of publication as it is essentially a proof-of-concept of a form of lab-scale turbulence generation that would be quite useful for the readership of this journal. I have divided my feedback below into comments on the work as a whole and small typographical or wording corrections.

COMMENTS:

1) The introduction is a little bit scant, particularly with respect to the literature review. While there is no minimum number of referenced works required for a paper, at 13 references (of which only 5 are full/non-conference journal papers), this is one of the works with the fewest references I have read in some time. As briefly pointed out in the introduction, active grids are likely the main competitor to this methodology but are hardly addressed in the text. Perhaps this is an area where more scope can be provided in terms of showing previous work or highlighting limitations of other methodologies; for instance, active grids struggle to create sudden changes such as those that can be made here. Numerous active grid papers address generating atmospheric/wind turbine conditions, e.g., Cekli & van de Water (Exp Fluids 2010), Knebel et al. (Exp Fluids 2011), Hearst & Ganapathisubramani (Wind Energ 2017), and active grids have been used in direct conjunction with model turbines in several studies as well, e.g., Cal et al. (J Renew & Sustain Energ 2010), Rockel et al. (Renew Energ 2017), Vinod & Banerjee (App Energ 2019), Li et al. (Renew Energ 2020). Moreover, limitations of using static grids can be made explicit, e.g., Devinant et al. (J Wind Eng Ind Aero 2002), Bartl et al. (Wind Energ Sci 2018). While I am not promoting that the authors must add these particular references, drawn from the top of my head, these or similar would certainly help provide context for the reader and multiple opportunities for comparison and contextualisation in the body of the text. The latter are somewhat lacking. Please

do not just add these specific references to a list that is already in the paper but instead use them as a starting point to expand the introduction slightly.

2) The authors use the term "gust" to describe the event created by the passing of the chopper blade. However, from the results, e.g., Fig 3, it appears these events are net losses in momentum, although there is a short increase at the start. To me this is a "lull", i.e., decrease in the wind speed, rather than a "gust", which is often associated with an increase. I thus suggest the authors either modify their terminology or include a definition for gust that includes the net momentum loss events shown here as "gusts".

3) Is 20 cycles sufficient for convergence of the results? Can some evidence be provided?

4) The stated frequency response of 38 kHz for a standard hot-wire seems quite high. Are the authors sure about this? See for instance Hutchins et al. (Exp Fluids 2015) where they suggest the true frequency response does not typically exceed ∼7 kHz.

5) In Figure 3a the red line has some non-zero thickness outside of the primary blade passing events. What does this signify?

6) I think some comments about the spectra are lacking a little bit of rigour. For instance:

a. Using the word "flat" to describe the low-frequency region is not exactly accurate. The discussion of the blade pass frequency and its harmonics is good though.

b. I am not sure there is in fact anything overly insightful about the spectral breakdown into the various components. This is well known, for instance, in the amplitude modulation community, c.f., the review by Dogan et al. (Fluid Dyn Res 2019).

c. There rigorously is no "f^-5/3" rule/law. This comes from Kolmogorov's analysis of the second-order structure function in isotropic turbulence. From dimensional analysis, in wavenumber space this results in k^-5/3, which is typically converted to a frequency spectrum by invoking Taylor's frozen flow hypothesis (e.g., Laizet et al. (Phys Fluids

2015)). So, while looking for fˆ-5/3 isn't wrong, it is a few steps removed from the theory. I only bring this up because no reference is made to where fˆ-5/3 comes from. Simply adding a sentence pointing in the direction of the source would be more rigorous.

7) With respect to the turbulence intensity used, this appears to be defined relative to a global incoming velocity rather than the local mean. What do these figures look like when normalized by the local mean? I am curious to see them in a reply to this comment; the authors do not necessarily need to include them in a revision at this point, but I will reserve final judgement on this.

8) I am not sure the breakdown into "inner" and "outer" scales (ln. 206) is used rigorously here either. These words mean specific things in most turbulent flows; for instance, for turbulence in general the inner variables scale the dissipation range (e.g., Kolmogorov variables) and the outer variables scale the energy containing range (e.g., the integral scale and the TKE). In wall-bounded flows, viscous based units (e.g., the viscosity and the friction velocity) are the inner units, while the boundary layer thickness, pipe radius, or channel height and the mean centerline/bulk/freestream velocity are the outer units. So, while I do not disagree with the authors' breakdown in principle, I am not sure the terminology used is the best, as it has specific meaning in other flows where the integral scale is an outer unit not an inner one. Moreover, the inner units of a turbulent flow are also still inner units here.

9) The discussion section actually acts mostly as a summary, and, as indicated above, I am not sure the discussion of the spectra in this section is in fact novel. If the spectral discussion is removed here, then this section can be merged with the conclusions.

10) This facility can produce many different flows, but only two are shown. Why is that? I would be interested in seeing more. I don't know if more need to be included in the present study (it would be stronger if there were more), but I would like to know why the authors stopped here.

MINOR CORRECTIONS:

1) (ln. 26) "extend" should be "extent".

2) (paragraph starting on ln. 30) The Wester et al. (2018) experiment is described as using the same configuration as Wei et al. (2019b), but the Wester et al. experiment actually preceded the Wei et al. experiment. Perhaps the words used or the order they are presented in should be changed. 3) (ln. 38) The Makita paper should be referenced by the last name.

4) There are quite a few times in the manuscript (e.g., the sentence starting on line 55) where a preposition, e.g., "in", is missing after "downstream" or "upstream". These should be found and corrected.

5) (ln. 99) Use of word "over" here is confusing. That terminology most often is used to mean "divided by" but that isn't what the authors mean.

6) (ln. 99) Word "of" is missing in front of "the chopper".

7) (ln. 124 and ln. 152) "the to the flow..." is some sort of typographical error that occurs more than once.

8) (ln. 154) "a low chopper frequency of the chopper", correct this.

9) (ln. 201) Repeated "into".

10) In general, the structure and writing style are good, but there are a few typos and missing articles and prepositions that confuse things. Please review the writing in general and perhaps get another set of eyes.

---

## Author Comment (AC1) · 25 Mar 2020

We have finalized the response to the reviewers and would like to thank them for their work that did help to improve the quality of our work. The attached answer (zip folder) contains two files and is structured as follows:
We answer each comment directly (answer in blue) in the file "Review_Answers_Chopper.pdf", with:

- Answer to reviewer #1 on p. 1-12

- Answer to reviewer #2 on p. 13 - 22

- Answer to reviewer #3 on p. 23-30

[Figure]

The changes in the manuscript that can be found in the file "Chopper_ChangesInManuscript.pdf" are also marked in blue.

Please also note the supplement to this comment:
https://www.wind-energ-sci-discuss.net/wes-2019-107/wes-2019-107-AC1-supplement.zip

---

## Author Response (AR1)

**Answers to the reviewer comments received for the article**
**First characterization of a new perturbation system for gust generation: The Chopper**

Ingrid Neunaber and Caroline Braud

March 25, 2020

**1 Anonymous Referee #1**

**General Comments:**

In this work, the authors introduce and characterize a new method for generating large gust disturbances in a wind tunnel using a rotating blade. Two gust frequencies were tested, and hot-wire measurements showed large gust amplitudes in both cases. The gust signals were decomposed into mean, disturbance, and turbulence components, and each of these components was analyzed in detail. Overall, the study is well motivated, designed, and documented, and the capability of the "chopper" to generate significant, repeatable gust events is conveyed convincingly. I would prefer, however, that the aerodynamics of the device be considered in more detail. Since only two gust frequencies are studied, a more comprehensive explanation of the physical principles that govern the gust generator's influence on the flow would greatly strengthen the characterization in this paper. Thus, while the gust-generation mechanism itself is novel and of interest to the wind-energy community, I believe a more detailed analysis of the results is required before this paper can be accepted for publication.

**Answer**
The authors would like to thank the reviewer who has suggested some very interesting and important points to address in this paper, especially with respect to the physical interpretation behind the gust generation process. We believe that the paper has been improved by including these aspects, and the detailed answers are given below.

**Specific Comments:**

**S1: [Section 1]** The introduction motivates the problem well, and the review of relevant literature is concise and effective. I note, however, that the EOG profiles in Figure 1 are never compared with the gust profiles induced by the "chopper" mechanism. The authors could make the connection between these classes of gust profiles clearer. It should also be specified that the gusts of interest to this study are in the stream-wise direction.

**Answer**

a) on the subject of *comparison with the EOG profiles:*

The gust produced by the chopper system is an inverse gust that cannot be compared directly to the EOG profiles, which are in turn also far from the shapes of real atmospheric gusts (for this reason, we also removed the EOG plot from the paper). However, main features like the gust characteristic time, the gust amplitude and the velocity fall times have been compared to the IEC EOG. This has been clarified in the revised article:

- **p. 3, ll. 89**: *"Further customization of the perturbation system will be needed to match realistic gusts like the one discussed for example by [3] which are far from the ideal gust shape provided by the IEC-61400-1 norm."*

- **p. 14, ll. 252**: *"The brief increase of the velocity that exceeds the inflow velocity is followed by a rapid velocity decrease ($18\,ms^{-1}$ in $2.5\,s$ which is significantly stronger than the decrease of $9.8\,ms^{-1}$ in $2.8\,s$ given as example for an EOG according to the IEC-61400-1 norm)"*

- **p. 15, ll. 275**: *"With a chopper blade width of $w_{CH} = 20\,cm$ as presented in this setup, the inverse gust amplitude is higher than proposed by the IEC-64100-1 norm. The changes in the velocity can be rapid, for example approximately $18\,ms^{-1}$ in $2.5\,s$ in case of $f_{CH1} = 0.04\,Hz$ and up to $25\,ms^{-1}$ in $0.25\,s$ in case of $f_{CH2} = 0.4\,Hz$."*

- **p. 16, ll. 290**: *"The approximate gust characteristic times are $\Delta t(f_{CH1}) \approx 10\,s$ and $\Delta t(f_{CH2}) \approx 0.9\,s$. $\Delta t(f_{CH1}) \approx 10\,s$ matches the characteristic time of the IEC EOG $\Delta t_{IEC} = 10.5\,s$ closely. $\Delta t(f_{CH2}) \approx 0.9\,s$ is scaled to wind tunnel dimensions: The ratio of chord length c and gust characteristic time should be constant which is achieved for the airfoil that we will use with $c = 0.09\,m$."*

b) on the subject of the *stream-wise direction of the gust*:

We agree with the reviewer and modified the introduction accordingly, including examples of transverse gust generations, and it has been clarified in the article that the gusts of interest in this study are in the stream-wise direction.

**S2: [Section 2]** More information about the closed-loop wind tunnel should be given. Since the test section of the wind tunnel is presumably discontinuous due to the construction of the gust generator, statistics regarding the baseline flow conditions in the measurement area would help isolate the influence of the gust generator from any background effects (especially in light of the turbulence intensities presented later in the

[Figure]

Figure 1: Turbulence intensity along the test section

work, e.g. Page 5, Line 94). The limitation of this gust-generation mechanism to wind tunnels with discontinuous test sections should also be stated, or modifications to the design for fully closed test sections should be proposed.

**Answer**

We agree with the reviewer that the discontinuity in the test section should be commented on. The chopper blade is passing through the test section and therefore there exists an open gap of approximately 8 cm between the chopper blade and the test section at the inlet. One of our first concerns after mounting the chopper system was whether the turbulence intensity would still be low when the chopper was not in use while the gap was left open. The turbulence intensity has been checked without movement of the chopper blade (the blade angle was $\beta = 90°$ and the blade was therefore not blocking any part of the inlet), and it was found to be below 0.3% along the test section (see figure 1). This low turbulence intensity is also retrieved during the operation of the chopper system between gusts (cf. p. 7 l. 160). Overall, we believe that the discontinuity in the test section does not induce background effects that alter the results.

A sentence has been added in the reviewed article to inform the reader about the level of the turbulent intensity outside the operation of the chopper system (**pp.4, ll.111**):

*"The turbulence intensity of the wind tunnel is below 0.3% throughout the test section, and it was verified prior to the experiments that the flow in the empty wind tunnel is homogeneous."*

**S3: [Section 2]** The design of the "chopper" system should be further motivated by aerodynamical considerations.

a) Is the primary purpose of the mechanism to produce large fluctuations in blockage, in order to create large oscillations in the free-stream velocity? If so, is there a

qualitative difference in the method to closing and opening an active grid?

b) Is the rotational motion of the blade intended to induce additional flow-normal velocity fluctuations due to the shearing action of the blade on the incoming flow?

c) If, as the authors point out in Figures 6 through 9, the resulting flows are asymmetric in the test section, why was a rotating system used instead of a rising and falling guillotine-like blade?

d) The authors should explain why this particular design was chosen in light of their goal of creating large-amplitude turbulent gusts.

**Answer**

a) The objective of the chopper system is to generate sudden velocity changes in combination with turbulence which are generally targeted separately in other perturbation devices. Active grids struggle to create sudden strong velocity changes such as those that can be made with the present chopper blade, which is now also highlighted in the revised manuscript.

b) The objective of the rotational motion of the blade is twofold: firstly, to generate repeatable and reproducible perturbations for stochastic analysis purposes; secondly, to generate rapid perturbations by increasing the rotational frequency, which is in addition found to induce a higher peak response of the stream-wise velocity. Those features are also expected within a real gust, and therefore, it is interesting to analyze the impact of it on blade or rotors dynamics.

c) There is indeed a transverse asymmetry in the mean flow field that is decreasing with the stream-wise distance. This transverse asymmetry is however not an important matter for at least three reasons:

  - there exists a downstream position from which this asymmetry is approximately negligible
  - often, the atmospheric flow has a transverse flow variation, and this setup can therefore be used to study the progressive influence of a transverse flow asymmetry on an airfoil
  - in the progress of customization of the chopper system, non-uniform chopper blades can compensate this asymmetry

To avoid complex mechanisms which will unavoidably slow down the process, such as a mechanism to pull up a guillotine-like blade, the choice of a rotating system was found ideal. The known drawback of a transverse asymmetry in the test section can be taken into easily account by designing a blade that is non-uniform in the span-wise direction by taking into account the radial blade speed variation due to rotation. For the proof of concept of this new perturbation system, this blade customization was not performed, and instead, an estimation of the span-wise non-uniformity with stream-wise distance has been preferred.

d) The chopper design was motivated by the excitation of high-frequent rotor or airfoil dynamics for control purposes. The design was inspired by one of the techniques commonly used to extract transfer functions of unsteady pressure sensors, namely the generation of a pressure step function using devices such as chock tubes or the balloon device (see e.g. [10]). The advantage of this method is that only one test of short characteristic time is sufficient to cover the entire frequency domain of interest. The basic idea was to generate a sudden, strong velocity change similar to a step function. The flow change needed to be at least as fast as the fastest phenomena encountered in the atmosphere (gust) when scaled to wind tunnel dimensions for a down-scaled airfoil, which leads to a step function at least as fast as 0.1s. In addition, turbulence should be superimposed onto the flow as turbulence is found to significantly modify the airfoil and rotor aerodynamics (see e.g. [5], [11] and [12]). With the proposed design, turbulence was expected to be automatically induced by the wake of the thick chopper blade with sharp edges. The customization of this background turbulence is for now left for further tests.

The authors agree with the reviewer that the motivation of this new perturbation system was unclear. However, details on techniques inspiring the chopper blade design or details on the motivation of this specific mechanism were found unnecessary and even detrimental for the clarity of the article.

We agree however that motivation for rotational motions and guidelines to avoid flow asymmetry in the test section should be added in the reviewed article:

**p. 3 ll. 77**: *"In this study, we will present a new perturbation system with a unique mechanism consisting of a rotating device. The objective of this new perturbation system, called the "chopper", is to combine sudden, strong mean flow changes and a turbulent background which is normally targeted separately in other perturbation devices. Targeted are the perturbations that cannot be accounted for with today's global control mechanisms and that will thus strongly affect the blade and rotor aerodynamics. This setup will therefore help to improve the insight into blade and rotor aerodynamics. Moreover, its application is also particularly interesting for the development of active flow control (ACF) devices. Using for example plasma actuators, a fast and local control with up to 10 kHz can be achieved, and up to a few kHz can be achieved in case of micro-jet actuators (see e.g. [?] and [6]). Those local flow control mechanism systems are suitable to alleviate loads from rapid and turbulent atmospheric perturbations and thus decrease fatigue loads."*

**p. 4, ll. 105**: *"With this design, rapid and strong velocity variations can be generated while turbulence is also included. In addition, this setup can be customized easily in the future by changing the chopper blade."* **p. 19 ll. 346**: *"Overall, a new system was presented that is capable of generating large, rapid velocity fluctuations while also inducing turbulence. By means of the downstream position and the chopper frequency, experiments can be executed in different regions of the flow field with different characteristics and different complexity that increases towards the inlet. For example, the aerodynamic behavior of an airfoil exposed to the flow can be studied downstream where the flow is already quite homogeneous, and then, it can be compared to the aerodynamic*

*behavior of the airfoil positioned farther upstream in the inhomogeneous region where three-dimensional effects will be present. This gives an opportunity to study the effect of radial inflow changes which may cause similar effects as rotational augmentation that is observed in rotational blades due to an additional radial flow (cf. [2]).*

*This investigation has shown us limits and opportunities of the setup and the measurements which helps us to improve the setup in the future. The currently very high blockage induced by the chopper blade will be reduced in the future by reducing the chopper blade width. We expect this to also decrease the currently very high gust amplitudes. Also, new chopper blade designs are possible that reduce the inhomogeneity of the flow field. To generate "normal" instead of inverse gusts, a design of a rotating disc with varying blockage is also possible. As the flow between the gusts has a low turbulence intensity, in addition, background turbulence can be added by installing a grid upstream of the chopper blade."*

**S4: [Section 2, Page 4, Line 78]** Why were 20 gust events chosen for these experiments? An analysis of statistical convergence would make the study more comprehensive.

**Answer**

We agree with the reviewer that the analysis of statistical convergence of 20 gust events was missing in the paper, and we have therefore added a plot in section 2, figure 2 on p. 6, and comment on it on p. 6, ll. 143. As can be seen from this plot, a good convergence is achieved at 20 gust events while still keeping measurement characteristic times at a reasonable level.

**S5: [Section 3.1]** Why is it that the higher rotation frequency leads to stronger gusts (Page 5, Lines 97-98)? Why do the gust velocities in this case exceed twice the inflow velocity? I highlight these observations as examples: the authors are very comprehensive in explaining their findings, but a physically motivated discussion of the reported phenomena is generally lacking. I would prefer that these observations be explained (and further analyzed as necessary), so that the generalizability of these findings can be inferred.

**Answer**

We agree that a physical discussion is certainly of interest for this new flow configuration. Therefore we did add explanations to certain features of the gusts in the paper in chapter 3.3 and hope that this will add to the comprehensibility:

**p. 12, ll. 234**: *"While the inverse gust generated with a chopper frequency of $f_{CH1} = 0.04\,Hz$ shows a smaller increase of the velocity in the beginning and is mainly characterized by a strong velocity deficit, in case of the faster chopper frequency, $f_{CH2} = 0.4\,Hz$, the increase in the beginning is much stronger and the velocity deficit afterwards less pronounced. The strong velocity increase can be explained by the faster blockage of the inlet in case of $f_{CH2}$: As the mass flow rate is expected to be similar, the faster blockage forces the flow to adapt quicker which leads to a sudden acceleration. Because the flow has more time to adapt in case of $f_{CH1}$, only little to no acceleration is present. For both chopper frequencies, the highly unsteady flow induces additionally strong fluctuations, and the combination of these strong fluctuations with the acceleration of the flow in case*

*of $f_{CH2}$ leads to those extreme velocity peaks."*

Figure 2 illustrates the superposition of the underlying gust shape and the turbulent fluctuations for one example (for this illustration, the gust was smoothed with a smaller bin than used in the paper). It can be seen that the amplitude of the smoothened gust (black) is approximately 25 m/s for $f_{CH1}$ and 35 m/s for $f_{CH2}$, and the fluctuations have an amplitude of up to 20 m/s respectively 15 m/s.

**S6: [Section 3.1, Page 5, Line 101]** Please comment on why an isotropic-turbulence assumption applies to this flow. Given the character of the forcing, I would expect that the generated turbulence would be significantly anisotropic.

**Answer**

Thank you for this critical question, in fact, Kolmogorov's theory of ideal turbulence is not expected to hold in this flow. We corrected the passage in the text on p.8, ll. 171.

**S7: [Section 3.1]** The decomposition of the gust into three components is helpful for the analysis of the results. Please specify the method by which the gust shape and turbulent fluctuations were isolated from the mean flow, as the method by which these were obtained affects the spectral signatures of the gust and fluctuation profiles. In Section 4, Page 14, Line 14, the authors apply a second-order filter to obtain the corresponding spectra. If this was the method used to achieve the initial decomposition, how was the cutoff frequency selected? Why wasn't a filter with a sharper roll-off used? These details are needed to explain whether the fluctuations in the gust profiles shown in Figures 11 and 12 are part of the mean gust profile or turbulent fluctuations that passed through the filter due to the relatively gentle roll-off of the second-order filter.

**Answer**

We appreciate your comment and clarified the text passage concerning the extraction of the mean gust shape and the fluctuations accordingly. It now reads:

*"The underlying inverse gust shape is here calculated by phase-averaging the gust events and smoothing with a moving average with a window size of $0.1$ s. The fluctuations are then extracted by subtracting the underlying inverse gust shape from the respective gust."*
(**p. 8, ll. 184**)

We only used a Butterworth filter for the demonstration of the decomposition in the spectral domain (figure 16(b) - first version of the manuscript) in the first version of the manuscript. In all other plots, the mean underlying gust shape is and was extracted by smoothing the phase-average over all gust events within the respective time series. As the use of two different methods may have been a bit irritating, we changed the method of calculating the spectrum of the recurring gusts: In the revised manuscript, we sample down the time series to $f_s = 10Hz$ by taking only every 10000th data point so that the spectrum contains frequencies up to $5Hz$. The frequency was chosen to be both in accordance with the window used for smoothing the mean gust shape and to give a spectrum of the low frequency range that has a bit of overlap with the spectrum of the turbulent fluctuations. The text now reads (**p. 9, ll. 189**):

*"To calculate $\tilde{E}(f)$, the time series was sampled down to $f'_s = 10$ Hz by taking every 10000th data point. We use this method instead of applying a filter to emphasize the two*

[Figure]

Figure 2: illustration of the superposition of an underlying gust shape with turbulent fluctuations to shed light on very high velocity fluctuations: for $f_{CH1}$ (left) and $f_{CH2}$ (right), one gust (upper row, plotted in light blue) can be split into an underlying shape (black) and turbulent fluctuations (blue; lower row)

*distinguishable parts the spectrum is made of, and the frequency was chosen to have a small overlap between $\tilde{E}(f)$ and $E'(f)$. In case of $E'(f)$, the spectra of $u'$ of all inverse gust events within the time series are calculated and afterwards, the spectra are averaged."*

**S8: [Section 3.2]** The authors do a good job characterizing the mean flow fields downstream of the gust generator. Could the authors comment on the extent to which the spanwise flow asymmetry shown in Figures 6 through 9 would affect wind-tunnel experiments (e.g. of nominally 2D wind-turbine blade sections)?

**Answer**

This is an interesting question as the span-wise flow asymmetry homogenizes with increasing downstream distance. Therefore, when performing aerodynamic experiments on a 2D airfoil, we can position the blade first far downstream where the span-wise variations are small and are thus expected to have little to no influence, and afterwards move the airfoil upstream to investigate the influence of the asymmetric flow. The asymmetry may cause similar effects as rotational augmentation that is observed in rotational blades due to an additional radial flow (cf. [2]), and this setup gives us the opportunity to study this impact. We included this in the outlook (**p.19, ll.347**):

*"By means of the downstream position and the chopper frequency, experiments can be executed in different regions of the flow field with different characteristics and different complexity that increases towards the inlet. For example, the aerodynamic behavior of an airfoil exposed to the flow can be studied downstream where the flow is already quite homogeneous, and then, it can be compared to the aerodynamic behavior of the airfoil positioned farther upstream in the inhomogeneous region where three-dimensional effects will be present. This gives an opportunity to study the effect of radial inflow changes which may cause similar effects as rotational augmentation that is observed in rotational blades due to an additional radial flow (cf. [2])."*

**S9: [Section 3.3]** Another instance of my request in S5, above. Please explain the qualitative trends described in the text. For example, why does the lower-frequency disturbance have a smaller increase in velocity at the beginning of the gust event than the higher-frequency disturbance (Page 9, Lines 149-150)?

**Answer**

We agree that a physical discussion is certainly of interest for this new flow configuration. Therefore we did add explanations to certain features of the gusts in the paper in chapter 3.3 and hope that this will add to the comprehensibility:

***p. 12, ll. 234***: *"While the inverse gust generated with a chopper frequency of $f_{CH1} = 0.04\,Hz$ shows a smaller increase of the velocity in the beginning and is mainly characterized by a strong velocity deficit, in case of the faster chopper frequency, $f_{CH2} = 0.4\,Hz$, the increase in the beginning is much stronger and the velocity deficit afterwards less pronounced. The strong velocity increase can be explained by the faster blockage of the inlet in case of $f_{CH2}$: As the mass flow rate is expected to be similar, the faster blockage forces the flow to adapt quicker which leads to a sudden acceleration. Because the flow has more time to adapt in case of $f_{CH1}$, only little to no acceleration is present. For both*

*chopper frequencies, the highly unsteady flow induces additionally strong fluctuations, and the combination of these strong fluctuations with the acceleration of the flow in case of $f_{CH2}$ leads to those extreme velocity peaks. In both cases, a small velocity increase around $t/T \approx 0.03$ is present."*

**S10: [Section 3.3]** In Figure 11b, there is a strong spike in the gust profiles just before $t/T = 0$ that is not present in the corresponding profiles for the low-frequency case. Is this significant, or is it just an artifact of filtering (see S7, above)? If it is significant, what caused it? Its presence seems to suggest that different physics are involved in the high-frequency case. These dynamics should be discussed if the effects of the gust generator are to be inferred outside of the two frequencies tested in the experiments.

**Answer**

The reviewer points out that a short increase of the velocity occurs around $t/T \approx 0.03$. While this peak is clearly more pronounced in case of $f_{CH2}$, a small peak is visible also at the same time in case of $f_{CH1}$.

On **p. 13, ll. 242**, we give an explanation for this peak:

*"In both cases, a small velocity increase around $t/T \approx 0.03$ is present. Its origin may lay in a flow change occurring when the chopper blade leaves the upper corner at the inner side of the inlet: The chopper blade rotates and progressively blocks the flow at the inlet of the test section. Once the blade has fully entered the inlet, the upper part of the test section is gradually unblocked, starting at the upper inner corner. The pressurized air is then abruptly released towards the low pressure area behind the chopper blade which leads to a sudden increase of the velocity as the flow is entrained into the wake of the blade before the flow adapts. Similar phenomena may happen when the chopper blade leaves the inlet, which is also marked by fluctuations of the velocity. As the flow has more time to adapt in case of $f_{CH1}$, the increase is less pronounced."*

This means that while three-dimensional effects will play a role, the underlying effect is universal with respect to the covered chopper frequency range.

**S11: [Section 3.3]** In Figure 13a, the data points between X/wCH = 6 and 9 appear to be offset uniformly downward from the trend of the rest of the data. Is this significant? Error bars for these data points would be helpful in answering this question, and in demonstrating the significance of the trends.

**Answer**

We agree with the reviewer that error bars are necessary, and we have therefore added them. As can be seen, the trend of the data points that the reviewer points out to be offset downward (figure 14a) is with respect to the error bars not significant. The variations are assigned to small variations in the chopper frequency between experiments, and improvements on the accuracy will be made in future experiments.

**S12: [Section 4]** The authors explain that the gusts have 'outer' and 'inner' scales, based respectively on the global gust disturbance and the integral length scale of the generated turbulence. It would be helpful to compare their relative magnitudes in physical space. Is there a frequency at which the outer scale would match the inner scale,

and if so, would the character of the gusts qualitatively change at this frequency?

**Answer**

The reviewer points out an interesting aspect. In the here presented results of the evolution of the integral length scale $L$ and the gust characteristic time $\Delta t$, it is shown that $L$ is in the order of magnitude of half the chopper blade width, and as known e.g. from results in the wake of a cylinder, the integral length is in the order of magnitude of the bluff body dimension. It is therefore expected that the integral length in the flow field would not significantly exceed the chopper blade width. In contrast, the gust characteristic time, that can be converted into a gust length by means of Taylor's hypothesis of frozen turbulence, depends on the chopper frequency and the inflow velocity. To generate global gust lengths of 20 cm would therefore imply either very high chopper frequencies (as $\Delta t$ decreases with increasing chopper frequency) or very low velocities, or a combination. For example, at 10 m/s, the gust characteristic time representing a structure of 20 cm would be 0.02 s. As the gust characteristic time is anti-proportional to the chopper frequency, the necessary chopper frequency would be expected to be around 20 Hz. As this scenario differs significantly from the here presented scenarios, we would expect different properties, and the interaction of these scales could add to this effect. However, the maximum chopper frequency that can be reached is 5Hz, and therefore we will not be able to investigate this interaction.

**S13: [Section 4, Page 14, Line 208]** Could the offset in the spectra be attributed in some way to the filtering method?

**Answer**

We do not believe that the offset can be attributed to the filtering method but rather that the normalization of the spectra with respect to the variance is the reason for the offset.

**Technical Corrections:**

T1: The use of the backslash to denote units in plots could be confusing. Parentheses or brackets would be preferable.

**Answer**

We appreciate your concern. However, as stated in the *rules and style conventions for expressing values of quantities* of the *International System of Units* (see e.g. `https://www.bipm.org/en/publications/si-brochure/section5-3.html`), "Symbols for units are treated as mathematical entities. In expressing the value of a quantity as the product of a numerical value and a unit, both the numerical value and the unit may be treated by the ordinary rules of algebra. This procedure is described as the use of quantity calculus, or the algebra of quantities. For example, the equation T = 293 K may equally be written T/K = 293."

We apply these conventions, but for an improved readability, we changed the axis labels to "$u/ms^{-1}$".

T2: Minor corrections to the references: 1) The two Wei et al. (2019) references on

Page 2, Line 31 are reversed. 2) The active-grid reference on Page 2, Line 38 should use the surname Makita, rather than the given name Hideharu. 3) The Wester et al. (2018) and Petrovic et al. (2019) citations require conference information.

**Answer**

Thank you, this has been addressed.

T3: The legend in Figure 5b may be clearer as "mean stream-wise velocity", since "gust velocity" carries the connotation of a disturbance rather than a mean quantity.

**Answer**

We have discussed this term and decided to not change this since it is introduced in the text and the term "mean stream-wise velocity" could also be mixed up with the mean velocity of the whole time series.

T4: The contours and labels in Figures 8b, 9b, and 15b are hard to read. Can they be shown in white instead?

**Answer**

Thank you, the labels in the dark parts of the plots have been changed to white.

T5: The caption to Figure 10 is written somewhat confusingly. It should read "The gust events are plotted in different colors, the smoothed average gust is marked in black, . . .". The color map of the gust events should also be specified, as it is in the caption to Figure 14.

**Answer**

Thank you, this has been addressed.

T6: There are a number of minor typographical and grammatical errors in the manuscript. They do not impact the legibility of the work (on the whole, it is very well written and logically constructed), but they should be addressed at some point.

**Answer**

Thank you, we have scrutinizer the article.

**2 Anonymous Referee #2**

**General**

A novel approach for generating high velocity fluctuations with a rotating chopper disc in a wind tunnel is presented. The effect of the disc is investigated for two different rotational velocities of the chopper at a constant wind speed. The absolute flow is measured detailed by a 1D hot wire array, which is traversed downstream. From this data the absolute velocity, the turbulence intensity and spectra are calculated. Furthermore, the author shows that each gust can be divided in the mean velocity, the gust shape and additional turbulence. Overall, the study shows a very interesting approach for a new device to generate velocity fluctuations under controllable conditions. The study has been carried out very conscientiously and to a large extent. In my opinion, however, the results show that the flow of the chopper needs to be investigated in more detail. The author shows just two different cases, which are compared to each other. Furthermore, the high turbulence intensities of up to 60%, as well as an absolute velocity minimum close to 0 m/s show that a hot wire is not a suitable measuring instrument for measurements near the chopper. The study would therefore benefit greatly from a measurement technique that can distinguish between positive and negative velocities. LDA measurements, for example, are suitable for this purpose. Also the used wind tunnel seems to have a huge problem with the high blockage induced by the chopper, which is indicated by the large recovery time of the velocity. The approach of flow modulation upstream presented here is very interesting and the results presented here show a nice first insight, but in my opinion the results must be looked at more closely and above all more critically due to the used hot wires and the limited cases.

**Answer**

The authors would like to thank the reviewer for this review that addresses important aspects especially with respect to the wind tunnel setup and the measurement technique. We did revise the paper and believe that these comments helped to improve the quality of the overall work. In addition, there are some remarks that are not addressed in the paper but that we would like to discuss in detail here.

The reviewer remarks that only two different cases are compared, and he points out the need for a more detailed investigation. We agree with the reviewer in that regard. However, we would also like to emphasize that this work is an initial investigation carried out to gain a first understanding of the chopper, i.e. the mean flow field, and the main gust features like shape, amplitude, characteristic time and turbulence, to customize the chopper prior to more detailed, improved experimental investigations in the future. The two investigated cases were chosen specifically for later applications, as will be explained below. We did add physical explanations of some flow characteristics in the revised manuscript and hope that this will help the readers and the reviewer.

The reviewer is concerned that the use of 1D hot-wire anemometry significantly alters the results. While the authors agree that the use of 1D hot-wire anemometry can be tricky and has to be handled with care, we would also like to highlight some aspects to

support the use of these sensors:

- The calibration has been carried out in a large range of velocities (0m/s - 55 m/s) to cover the whole range of velocities present in the measurements

- Histograms of the velocities have been looked at randomly to verify that the distributions are reasonable and do not show any anomalies caused by backflows that would have been captured as positive flow events

- The turbulence intensity presented in the manuscript is by definition a global one (this is also pointed out by reviewer 3). Therefore, the standard deviation includes both the gust shape and the fluctuations, and this leads to a very high turbulence intensity. When calculating the local turbulence intensity over for example bins of 500 samples, the maximum turbulence intensity is 30% and the average turbulence intensity is below 15%. While these values are still high, they are significantly lower than the global turbulence intensity

- one of the main advantages of hot-wires is the high temporal resolution, and to characterize the unsteady flow, this is important

In addition, we would like to refer again to the purpose of this work as pioneering study of the system.

The reviewer also raises concerns about the blockage that is induced in the wind tunnel. We agree that the blockage is very large in this initial setup. This study also helps to improve the setup with respect to the blockage since smaller blades could be used or a one-armed blade.

In the following, we will address the specific comments in detail.

**Specific Comments:**

**1. Section 1**: The goal of the study is well motivated, as it is a current problem of the wind energy community. Also the current literature is well reviewed. However, the work of D. Greenblatt (e.g. Unsteady Low-Speed Wind Tunnels (2016)) should also be included in the literature, as here speed variations due blockage using a louver have already been worked with.

**Answer**

Thank you for pointing out this interesting work of a downstream blockage system. We reviewed the introduction and added this work together with other approaches to generate gusts.

**2. Section 1**: The authors show in figure 1 the EOG which serves as motivation. Also, a characteristic time of 10.5s is mentioned. Unfortunately, the authors do not compare their results of the chopper against those defined gusts. I am also missing the mentioned frequency of the gust in the presented results, or an explanation why this frequency was not used.

**Answer**

We agree with the reviewer that a comparison of the EOG with the gust generated by the chopper was missing. The gust produced by the chopper system is an inverse gust that cannot be compared directly to the EOG profiles, which are in turn also far from the shapes of real atmospheric gusts (for this reason, we also removed the plot from the paper). However, main features like the gust characteristic time, the gust amplitude and the velocity fall times have been compared to the IEC EOG. In fact, the two chopper frequencies were chosen to generate gusts that would a) match the characteristic time of the IEC EOG and b) have a characteristic time that is scaled down to wind tunnel dimensions (i.e. typical blade chord of 0.1m) in order to have a similar ratio of gust characteristic time over chord length that is encountered at full scale, ratio 10s/1m, which leads to a gust characteristic time of around 1s. This has been clarified in the revised article:

- **p. 5, ll. 137**: *"Two chopper frequencies, $f_{CH1} = 0.04\,Hz$ and $f_{CH2} = 0.4\,Hz$, have been investigated. The frequencies were chosen to generate a) inverse gusts with a characteristic time similar to the gust characteristic time $\Delta t_{IEC} = 10.5\,s$ used in the IEC-61400-1 norm, and b) inverse gusts with a characteristic time that is scaled down to wind tunnel dimensions. For this, the ratio of gust characteristic time over chord length $\Delta t/c$ is kept constant, which yields $\Delta t \approx 1\,s$ for a blade chord length of $c = 0.1\,m$ and thus $\approx 1\,s/0.1\,m$ in the wind tunnel as compared to $\approx 10\,s/1\,m$ for full scale measurements."*

- **p. 14, ll. 252**: *"The brief increase of the velocity that exceeds the inflow velocity is followed by a rapid velocity decrease ($18\,ms^{-1}$ in $2.5\,s$ which is significantly stronger than the decrease of $9.8\,ms^{-1}$ in $2.8\,s$ given as example for an EOG according to the IEC-61400-1 norm)."*

- **p. 15, ll. 275**: *"With a chopper blade width of $w_{CH} = 20\,cm$ as presented in this setup, the inverse gust amplitude is higher than proposed by the IEC-64100-1 norm. The changes in the velocity can be rapid, for example approximately $18\,ms^{-1}$ in $2.5\,s$ in case of $f_{CH1} = 0.04\,Hz$ and up to $25\,ms^{-1}$ in $0.25\,s$ in case of $f_{CH2} = 0.4\,Hz$."*

- **p. 16, ll. 290**: *"The approximate gust characteristic times are $\Delta t(f_{CH1}) \approx 10\,s$ and $\Delta t(f_{CH2}) \approx 0.9\,s$. $\Delta t(f_{CH1}) \approx 10\,s$ matches the characteristic time of the IEC EOG $\Delta t_{IEC} = 10.5\,s$ closely. $\Delta t(f_{CH2}) \approx 0.9\,s$ is scaled to wind tunnel dimensions: The ratio of chord length $c$ and gust characteristic time should be constant which is achieved for the airfoil that we will use with $c = 0.09\,m$."*

From this additional explanation, we hope that the link between the IEC EOG and the chosen chopper frequencies is now clearer.

**3. Section 2**: The author should comment on the use of 1D hot wires since the wake of a flat plate is a very complex and three-dimensional flow.
**Answer**

We agree with the reviewer that 1D hot-wire measurements are not suitable for a full documentation of this three-dimensional flow. However, it is sufficient to give a first estimation of

- the mean flow organization

- the time response of the flow (gust characteristic time/acceleration/deceleration)

- first description of the turbulence

- the downstream evolution of these quantities

which was targeted here to help in future customizations of the chopper (see recommendations for future customization of the chopper system). The authors would like to point out that the most important parameters of this new perturbation system are in the time domain, and we therefore need a very high temporal resolution of the measurements which is precisely what the 1D hot-wire is capable of doing. We added a comment in the paper: **p. 5, ll. 127**:

*"We are aware that the wake of the rotating chopper blade is complex and three-dimensional which may limit the use of 1D hot-wire anemometry. However, the scope of this study is the investigation of the evolution of the mean velocity, inverse gust shape and the turbulence characteristics throughout the test section. As we scan the flow field with several probes and at multiple positions, arriving at 65 measurement points in total, a spatial three-dimensional characterization at a very high sampling frequency is achieved that allows us the detailed investigation of these quantities and also the rapid flow reactions and the high frequent, small-scale turbulence."*

In addition, we would like to bring back into mind that we have been as discussed above carefully looking at the data and the calibrations to make sure the data is (within the limitations of 1D hot-wire anemometry) reliable.

**4. Section 2 (P4 L76)**: The authors do not comment why the flow velocity of 25 m/s is used and why they choose the presented frequencies. They also choose to measure "more than 20 gust events" and do not comment why they choose to do so. In addition for the 0.4 Hz case the number of measured gusts is doubled compared to the 0.04Hz case. Were all data used for the evaluation?

**Answer**

We agree with the reviewer, that the mentioned aspects have not been clear in the manuscript.

- In this investigation, an inflow velocity of 25 m/s is used as this will be the velocity used later in aerodynamic experiments. The passage has been updated in the manuscript, **p. 5, ll. 135**:
  *"The inflow velocity was $u_0 = 25\ ms^{-1}$, the velocity we will use for aerodynamic measurements of an airfoil in the future."*

- As discussed above, the chopper frequencies were chosen to generate gusts matching the characteristic time of th IEC EOG in full scale and downscaled to wind tunnel dimensions. We did update the manuscript, **p. 5, ll. 134:**
  *"Two chopper frequencies, $f_{CH1} = 0.04\ Hz$ and $f_{CH2} = 0.4\ Hz$, have been investigated. The frequencies were chosen to generate a) inverse gusts with a characteristic time similar to the gust characteristic time $\Delta t_{IEC} = 10.5\ s$ used in the IEC-61400-1 norm, and b) inverse gusts with a characteristic time that is scaled down to wind tunnel dimensions. For this, the ratio of gust characteristic time over chord length $\Delta t/c$ is kept constant, which yields $\Delta t \approx 1\ s$ for a blade chord length of $c = 0.1\ m$ and thus $\approx 1\ s/0.1\ m$ in the wind tunnel as compared to $\approx 10\ s/1\ m$ for full scale measurements."*

- More than 20 gust events have been chosen as statistical convergence is approximately reached at 20 gust events. In the paper, we have added a plot that shows this in section 2, figure 2 on p. 6, and comment on it on **p. 6, ll. 143**. As can be seen from this plot, a good convergence is achieved at 20 gust events while still keeping measurement characteristic times at a reasonable level.
  *"To measure a sufficient amount of inverse gust events for each chopper frequency, the measurement times were adapted and are $t_1 = 600\ s$ for $f_{CH1} = 0.04\ Hz$ and $t_2 = 120\ s$ for $f_{CH2} = 0.4\ Hz$, respectively. It was verified that the number of inverse gusts captured is sufficient to reach statistical convergence, which is achieved when capturing at least 20 inverse gust events as illustrated exemplarily in figure 2 for various points of time t during the respective gust event."*

**5. Section 3 (P5 Fig 3)**: The data show a large recovery time of the flow velocity after the chopper leaves the measurement section. This is not discussed by the authors but shows that the wind tunnel is probably not suited for such high blockages.
**Answer**
We agree with the reviewer that the present blockage is certainly too high for the wind tunnel which is indicated by the slow recovery time after a gust that keeps the flow from fully recovering in case of the high chopper frequency. The presented pioneering work makes this problematic visible and helps to customize the system in the future. To optimize the blockage and include the recovery time, two measures are suggested:

- The blockage will be reduced by reducing the chopper blade width, **p. 19, ll. 355**:
  *"The currently very high blockage induced by the chopper blade will be reduced in the future by reducing the chopper blade width."*

- The recovery time can be taken into account by using one-armed chopper blades in case of higher chopper frequencies, **pp. 7, ll. 163**:
  *"This indicates that the recovery time of the mean velocity to $u_0$ between the gusts needs to be taken into account in future experiments. One possibility to avoid the gusts to interact with each other at high chopper frequencies would be to change*

*the duty cycle, i.e. to use a chopper blade with only one arm."*

**6. Section 3.1 (P5 L97)**: For the case with higher rotation higher fluctuations are measured. The duplication of velocity do not match with the induced blockage. This indicates strong aerodynamic effects of the chopper since the measured spikes occur when the chopper enters the test section. I think the authors need to be careful with interpreting these peaks without knowledge of the ongoing effects.

**Answer**

We agree that the spikes are striking, but we found that the spikes are not related with the induced blockage and therefore are not expected to match with it. As explained below, the extreme spikes occur due to a superposition of the mean gust shape with fluctuations. We have looked closely into the data and we came up with a physical explanation that we also added in the manuscript:

**p. 12, ll. 234**: *"While the inverse gust generated with a chopper frequency of $f_{CH1} = 0.04\,Hz$ shows a smaller increase of the velocity in the beginning and is mainly characterized by a strong velocity deficit, in case of the faster chopper frequency, $f_{CH2} = 0.4\,Hz$, the increase in the beginning is much stronger and the velocity deficit afterwards less pronounced. The strong velocity increase can be explained by the faster blockage of the inlet in case of $f_{CH2}$: As the mass flow rate is expected to be similar, the faster blockage forces the flow to adapt quicker which leads to a sudden acceleration. Because the flow has more time to adapt in case of $f_{CH1}$, only little to no acceleration is present. For both chopper frequencies, the highly unsteady flow induces additionally strong fluctuations, and the combination of these strong fluctuations with the acceleration of the flow in case of $f_{CH2}$ leads to those extreme velocity peaks."*

Figure 3 illustrates the superposition of the underlying gust shape and the turbulent fluctuations for one example (for this illustration, the gust was smoothed with a smaller bin than used in the paper). It can be seen that the amplitude of the smoothened gust (black) is approximately 25 m/s for $f_{CH1}$ and 35 m/s for $f_{CH2}$, and the fluctuations have an amplitude of up to 20 m/s respectively 15 m/s.

Still, additional aerodynamic effects that are not discussed here may add to the results. However, we would also like to emphasize that the present paper is a proof of concept of this new chopper system, and the main purpose of this work is to get an idea of the mean flow field and the gust shape, amplitude and characteristic time.

**7. Section 3.1 (P6 L103)**: The decomposition of the gust into mean velocity, fluctuations and turbulence is a nice approach. It should be complained in more detail, how the decomposition was calculated.

**Answer**

The authors would like to thank the reviewer for this comment. This decomposition approach is a quite standard approach when dealing with turbulent free shear flows with large-scale structures such as jets or wakes. The approach has been adapted to the present flow configuration. Details on how the decomposition is performed have been added on **pp. 8, ll. 180**:

[Figure]

Figure 3: illustration of the superposition of an underlying gust shape with turbulent fluctuations to shed light on very high velocity fluctuations: for $f_{CH1}$ (left) and $f_{CH2}$ (right), one gust (upper row, plotted in light blue) can be split into an underlying shape (black) and turbulent fluctuations (blue; lower row)

*"The investigation of the gust time series and the energy spectral density shows how the recurring inverse gust generated by the chopper can be decomposed into three parts: the mean velocity of the inverse gust, the underlying shape of the inverse gust as recurring global large-scale flow structure, and the turbulence within the gust. This triple decomposition is illustrated in figure 5. The inverse gust (figure 5(a)) has a mean gust velocity $\bar{u}$ (figure 5(b)) with an underlying inverse gust shape $\tilde{u}$ (figure 5(c)) with strong turbulent fluctuations $u'$ (figure 5(d)), and $u_G = \bar{u} + \tilde{u} + u'$. The underlying inverse gust shape is here calculated by phase-averaging the gust events and smoothing with a moving average with a window size of $0.1\ s$. The fluctuations are then extracted by subtracting the underlying inverse gust shape from the respective gust."*

**8. Section 3.2 (Fig 6 7)**: The flow field has been studied very thoroughly and the results are well presented. The results show nevertheless a high asymmetric shape. This seems to be correlated with the shape of the chopper and with the time the blade stays in the inlet of the wind tunnel. Are there plans to improve this with a different geometric chopper shape? How should one deal with such high asymmetries during experiments with model wind turbines or blades?

**Answer**

The asymmetries are indeed correlated with the shape of the chopper blade which is straight with a tip that is going faster than its root in the test section. We knew this from the start and we deliberately chose to keep the blade design as simple as possible to observe how the inhomogeneity eventually levels out with increasing stream-wise distance. We actually observe that the flow field is approximately homogeneous at the horizontal centerline from $8.5\,w_{CH}$ downstream. From our point of view, the asymmetry is not a problem but rather an opportunity to study the effect of either a radial inflow change on airfoils or the impact of sheared inflows on model rotors when using different stream-wise positions in the test section. This was also added in the manuscript (**p. 19, ll. 347**),

*"By means of the downstream position and the chopper frequency, experiments can be executed in different regions of the flow field with different characteristics and different complexity that increases towards the inlet. For example, the aerodynamic behavior of an airfoil exposed to the flow can be studied downstream where the flow is already quite homogeneous, and then, it can be compared to the aerodynamic behavior of the airfoil positioned farther upstream in the inhomogeneous region where three-dimensional effects will be present. This gives an opportunity to study the effect of radial inflow changes which may cause similar effects as rotational augmentation that is observed in rotational blades due to an additional radial flow (cf. [2])."*

To induce perturbations with transverse homogeneity, one possibility is to specifically change the shape of the blade for a certain chopper frequency so that a transversally homogeneous perturbation is induced. Another approach to generate "real" gusts instead of inverse gusts is to use a rotating disc with varying blockage, and in such a setup, homogeneity can also be accounted for. We briefly added this in the revised manuscript (**p. 19, ll. 356**),

*"Also, new chopper blade designs are possible that reduce the inhomogeneity of the flow*

*field. To generate "normal" instead of inverse gusts, a design of a rotating disc with varying blockage is also possible."*

**9. Section 3.3 (P10 Fig 10)**: The shown fluctuations seem to be cut at low velocities. Here it needs to be verified if the results are correct or if this is a result of the used 1D hot wire and the used calibration.
**Answer**
We appreciate the reviewer's concern about the measurements and would like to comment on the two points that are mentioned to lessen these concerns:

- The calibration has been carried out in a large range of velocities (0m/s - 55 m/s) to cover the whole range of velocities present in the measurements. The calibrations did not show any anomalies. It should be added that the data was calibrated directly by us and not by using additional software that also sometimes alters the data.

- Histograms of the velocities have been looked at randomly to verify that the distributions are reasonable and do not show any anomalies caused by backflows that were captured as positive flow events. Also the range of captured velocities was checked and while some flow events have low velocities ($>0.4$ m/s), they are never actually 0.

**10. Section 3.3 (P10 Fig 10)**: The average gusts show for both cases a peak at t/T = -0.05. The author does not comment on this peak, but it seems to be present in all gusts, since it shows up in the averaged data. The peak is also present for different positions as it can be observed in Fig. 11 as well.
**Answer**
We agree with the reviewer and updated the manuscript (**p. 13, ll. 242**), it now reads:
*"In both cases, a small velocity increase around $t/T \approx 0.03$ is present. Its origin may lay in a flow change occurring when the chopper blade leaves the upper corner at the inner side of the inlet: The chopper blade rotates and progressively blocks the flow at the inlet of the test section. Once the blade has fully entered the inlet, the upper part of the test section is gradually unblocked, starting at the upper inner corner. The pressurized air is then abruptly released towards the low pressure area behind the chopper blade which leads to a sudden increase of the velocity as the flow is entrained into the wake of the blade before the flow adapts. Similar phenomena may happen when the chopper blade leaves the inlet, which is also marked by fluctuations of the velocity. As the flow has more time to adapt in case of $f_{CH1}$, the increase is less pronounced."*

**11. Section 3.3 (P11 Fig 13)**: It is explained how the gust characteristic time is determined; however, I am missing error bars in the plot. Also an explanation why the thresh- old of 0.025 is used is not explained further, but is crucial for the calculated gust characteristic time.
**Answer**
We agree with the reviewer that error bars have been missing on this plot. We added

them and also discuss the choice of the threshold and hope that this clarifies things. The revised passage now reads (**p. 15, ll. 280**):

*"The gust characteristic time is calculated by taking the absolute value of the normalized gust fluctuations $u'/\bar{u}$ to find the interval in which the fluctuations frequently exceed a threshold set to be 0.025. This threshold was chosen to be low enough to capture the beginning of the gust well while also being high enough to neglect possible unexpected dynamics in the laminar part between the gust. In addition, the thresholds 0.02 and 0.03 were used to determine errors from the sensitivity of this method towards different thresholds. The results are plotted in form of error bars that may be asymmetric."*

**12. Section 3.4** : The integral length scales are calculated from the data, but is the information one could draw from this? I think if those numbers are presented the should be put in some context.

**Answer**

The authors would like to thank the reviewer for this suggestion. We have added a sentence in the discussion of the integral length scale (**p. 18, ll. 319**):

*"The integral length scale is of interest when carrying out aerodynamic experiments because the interaction between the airfoil and the flow structures depends on the ratio $L/c$. It is for example indicated in [4] that a smaller integral length in the flow delays stall."*

**3 Anonymous Referee #3**

This work describes the construction and output of a, to my knowledge, novel turbulence generator in a wind tunnel. This device is called "the chopper" and is essentially a spinning blade that cuts through, or "chops", the wind tunnel inlet. The blade can be spun at different frequencies, resulting in different intermittently turbulent flows in the wind tunnel test-section. One-component hot-wire measurements were performed in order to assess the chopper's output. It is demonstrated that the chopper imparts sudden changes on the centerline mean velocity, which is particularly useful as most other turbulence generating techniques cannot create a flow which responds so rapidly. Homogeneity and spectra are also investigated, demonstrating that the flow becomes more homogeneous as it develops downstream and that the rotational rate of the chopper blade and its harmonics are present at the low frequencies in the spectra, with a turbulent looking spectrum at higher frequencies. Both of these observations are perhaps obvious, but nonetheless are shown for the first time here for this novel setup. One feature of the paper I particularly liked was that it often made me ask a question in my head, and I found that the answer was present later in the paper. I think after some review, this work will be worthy of publication as it is essentially a proof-of-concept of a form of lab-scale turbulence generation that would be quite useful for the readership of this journal. I have divided my feedback below into comments on the work as a whole and small typographical or wording corrections.

**Answer**

We would like to thank the reviewer for this thorough discussion of our work, and we are sure that addressing these comments helped to improve our work. We will focus on the specific comments in the following.

**COMMENTS**

**1)** The introduction is a little bit scant, particularly with respect to the literature review. While there is no minimum number of referenced works required for a paper, at 13 references (of which only 5 are full/non-conference journal papers), this is one of the works with the fewest references I have read in some time. As briefly pointed out in the introduction, active grids are likely the main competitor to this methodology but are hardly addressed in the text. Perhaps this is an area where more scope can be provided in terms of showing previous work or highlighting limitations of other method- ologies; for instance, active grids struggle to create sudden changes such as those that can be made here. Numerous active grid papers address generating atmospheric/wind turbine conditions, e.g., Cekli & van de Water (Exp Fluids 2010), Knebel et al. (Exp Fluids 2011), Hearst & Ganapathisubramani (Wind Energ 2017), and active grids have been used in direct conjunction with model turbines in several studies as well, e.g., Cal et al. (J Renew & Sustain Energ 2010), Rockel et al. (Renew Energ 2017), Vinod & Banerjee (App Energ 2019), Li et al. (Renew Energ 2020). Moreover, limitations of us- ing static grids can be made explicit, e.g., Devinant et al. (J Wind Eng Ind Aero 2002), Bartl et

al. (Wind Energ Sci 2018). While I am not promoting that the authors must add these particular references, drawn from the top of my head, these or similar would certainly help provide context for the reader and multiple opportunities for comparison and contextualisation in the body of the text. The latter are somewhat lacking. Please do not just add these specific references to a list that is already in the paper but instead use them as a starting point to expand the introduction slightly.

**Answer**

The authors would like to thank the reviewer for the suggestions to improve the introduction. We have rewritten the introduction, adding more information and examples on turbulence generation in general and including more different setups used to generate gusts.

**2)** The authors use the term "gust" to describe the event created by the passing of the chopper blade. However, from the results, e.g., Fig 3, it appears these events are net losses in momentum, although there is a short increase at the start. To me this is a "lull", i.e., decrease in the wind speed, rather than a "gust", which is often associated with an increase. I thus suggest the authors either modify their terminology or include a definition for gust that includes the net momentum loss events shown here as "gusts".

**Answer**

The authors agree with the reviewer that the term gust might be misleading, and we therefore changed the terminology to "inverse gust" throughout the text.

**3)** Is 20 cycles sufficient for convergence of the results? Can some evidence be provided?

**Answer**

The reviewer points out an important question. More than 20 gust events have been chosen as statistical convergence is approximately reached at 20 gust events. In the paper, we have added a plot that shows this in section 2, figure 2 on p. 6, and comment on it on **p. 6, ll. 143**. As can be seen from this plot, a good convergence is achieved at 20 gust events while still keeping measurement characteristic times at a reasonable level.

**p. 6, ll. 143**: *"To measure a sufficient amount of inverse gust events for each chopper frequency, the measurement times were adapted and are $t_1 = 600\,s$ for $f_{CH1} = 0.04\,Hz$ and $t_2 = 120\,s$ for $f_{CH2} = 0.4\,Hz$, respectively. It was verified that the number of inverse gusts captured is sufficient to reach statistical convergence, which is achieved when capturing at least 20 inverse gust events as illustrated exemplarily in figure 2 for various points of time t during the respective gust event."*

**4)** The stated frequency response of 38 kHz for a standard hot-wire seems quite high. Are the authors sure about this? See for instance Hutchins et al. (Exp Fluids 2015) where they suggest the true frequency response does not typically exceed ∼7 kHz.

**Answer**

We would like to thank the reviewer for this remark. As we only tested the system by means of a square wave test, we changed the respective passage, **p. 4, ll. 120**, to

**5)** In Figure 3a the red line has some non-zero thickness outside of the primary blade passing events. What does this signify?

**Answer**

Between the gusts, the chopper signal from the induction sensor is noisy which explains the non-zero thickness, but since the blade passing is clearly identifiable this does not pose a problem.

**6)** I think some comments about the spectra are lacking a little bit of rigour. For instance:

a) Using the word "flat" to describe the low-frequency region is not exactly accurate. The discussion of the blade pass frequency and its harmonics is good though.

b) I am not sure there is in fact anything overly insightful about the spectral breakdown into the various components. This is well known, for instance, in the amplitude modulation community, c.f., the review by Dogan et al. (Fluid Dyn Res 2019).

c) There rigorously is no "f^-5/3" rule/law. This comes from Kolmogorov's analysis of the second-order structure function in isotropic turbulence. From dimensional analysis, in wavenumber space this results in k^-5/3, which is typically converted to a frequency spectrum by invoking Taylor's frozen flow hypothesis (e.g., Laizet et al. (Phys Fluids 2015)). So, while looking for f^-5/3 isn't wrong, it is a few steps removed from the theory. I only bring this up because no reference is made to where f^-5/3 comes from. Simply adding a sentence pointing in the direction of the source would be more rigorous.

**Answer**

The authors would like to thank the reviewer for these valuable comments concerning the discussion of energy spectra, and we will address the points in the following:

a) We updated the manuscript, it now reads (**p. 7, ll. 169**)
   *"At low frequencies, the spectrum is approximately constant but superimposed with the peak of the chopper frequency and its harmonics."*

b) We appreciate your skepticism regarding the added value the spectral decomposition might bring, and we are also aware that we are not the first ones to apply this method. As we wanted to nevertheless include this to show the reader how the turbulence is completely separated from the inverse gust shape, we added the plot to the discussion of the decomposition in section 3.1 while also following your suggestion in comment 9) to remove the discussion section (see below).

c) We agree with the reviewer that the discussion of the -5/3 decay of the energy spectra was lacking information while also wrongly stating that we find Kolmogorov's theory to be applicable. We therefore updated and corrected the passage on **p. 7, ll. 169**:

*" At low frequencies, the spectrum is approximately constant but superimposed with the peak of the chopper frequency and its harmonics. From $f \approx 10$ Hz, the energy spectral density decays with increasing frequency according to $E(f) \propto f^{-5/3}$ up to $f \approx 1000$ Hz. Kolmogorov predicted a scaling of the wave-number energy spectrum $E(k)$ according to $k^{-5/3}$ for "ideal" turbulence with dissipation in equilibrium (cf. [7], [8]). By means of Taylor's hypothesis of frozen turbulence, the wave-number spectrum can be converted to a spectrum in the frequency domain and the scaling becomes $f^{-5/3}$ (e.g. [9]). At first glance, it is therefore tempting to interpret the scaling found in the presented energy spectra as an indication of the existence of features of homogeneous isotropic turbulence. However, as the measurements that are presented here are carried out in the highly turbulent near wake of the chopper blade, Kolmogorov's theory is not expected to hold. Other studies such as the one presented by [1] find a similar behavior of the turbulence. The origin of the $-5/3$ scaling may therefore be the intermittency of the turbulent flow, as for example [9] and [13] discuss."*

**7)** With respect to the turbulence intensity used, this appears to be defined relative to a global incoming velocity rather than the local mean. What do these figures look like when normalized by the local mean? I am curious to see them in a reply to this comment; the authors do not necessarily need to include them in a revision at this point, but I will reserve final judgement on this.

**Answer**

We would like to thank the reviewer for this very interesting question. It is true that the turbulence intensity as presented in the paper is calculated with respect to the global inverse gust properties. We did emphasize this in the paper now. Also, as requested, in figure 4, we show examples of the calculation of a local turbulence intensity in the gust (row 1) that is calculated by dividing the gust into 50, 500 and 1000 bins and calculating the standard deviation and the mean velocity for each bin. In addition, the average local turbulence intensities of the gust are marked. The evolution of the mean velocity is plotted in the middle row for the three bin numbers, and the downstream evolution of the average local turbulence intensity calculated for a bin number of 500 is plotted as interpolated contour plot in the third row. It can be seen (row 1) that the local turbulence intensity strongly depends on the bin number which is expected because a small bin size (and thus a large bin number) does not include the strong gradients in the velocity change that cause large standard deviations. The values are significantly lower than the global turbulence intensity. When comparing the results for $f_{CH1}$ and $f_{CH2}$, the values are significantly higher for $f_{CH1}$. However, since the characteristic time of the gust generated in case of $f_{CH2}$ is a factor 10 smaller, it is also interesting to compare the case "500 bins for $f_{CH1}$" with the case "50 bins for $f_{CH2}$" which gives a similar number

of points in each bin, and here, the values are similar. Interesting is also that the typical wake shape with two peaks at the beginning and end of the gust that mark the shear layers is found although the data is collected with a fixed sensor and a moving obstacle.

**8)** I am not sure the breakdown into "inner" and "outer" scales (ln. 206) is used rigorously here either. These words mean specific things in most turbulent flows; for instance, for turbulence in general the inner variables scale the dissipation range (e.g., Kolmogorov variables) and the outer variables scale the energy containing range (e.g., the integral scale and the TKE). In wall-bounded flows, viscous based units (e.g., the viscosity and the friction velocity) are the inner units, while the boundary layer thickness, pipe radius, or channel height and the mean centerline/bulk/freestream velocity are the outer units. So, while I do not disagree with the authors' breakdown in principle, I am not sure the terminology used is the best, as it has specific meaning in other flows where the integral scale is an outer unit not an inner one. Moreover, the inner units of a turbulent flow are also still inner units here.

**Answer**
We would like to thank the reviewer for pointing this out, and we have changed the terminology to 'global scale' for the gust shape and 'outer turbulence scale' for the integral length.

**9)** The discussion section actually acts mostly as a summary, and, as indicated above, I am not sure the discussion of the spectra in this section is in fact novel. If the spectral discussion is removed here, then this section can be merged with the conclusions.

**Answer**
We agree with the reviewer on the discussion section which has therefore been merged with the conclusion as suggested. Regarding the spectral discussion, we believe it is interesting to show this decomposition, even though it may be trivial. As above-mentioned, it has however been displaced and can now be found after the flow decomposition in section 3.1 (see **pp. 8, ll. 186**).

**10)** This facility can produce many different flows, but only two are shown. Why is that? I would be interested in seeing more. I don't know if more need to be included in the present study (it would be stronger if there were more), but I would like to know why the authors stopped here.

**Answer**
The two flow cases were chosen to produce gusts with a characteristic time matching the one of the IEC EOG, and downscaled to wind tunnel dimensions. This information was missing in the first version but has been added in the revised manuscript:
**p. 5, ll. 136**: *"Two chopper frequencies, $f_{CH1} = 0.04\,Hz$ and $f_{CH2} = 0.4\,Hz$, have been investigated. The frequencies were chosen to generate a) inverse gusts with a characteristic time similar to the gust characteristic time $\Delta t_{IEC} = 10.5\,s$ used in the IEC-61400-1 norm, and b) inverse gusts with a characteristic time that is scaled down to wind tunnel dimensions. For this, the ratio of gust characteristic time over chord length $\Delta t/c$ is kept constant, which yields $\Delta t \approx 1\,s$ for a blade chord length of $c = 0.1\,m$ and thus $\approx 1\,s/0.1\,m$*

[Figure]

Figure 4: Calculation of the local turbulence intensity for the two chopper frequencies $f_{CH1}$ (left) and $f_{CH2}$ (right): In row 1, the local turbulence intensities are plotted for one exemplary gust at $X = 8.7w_{CH}$ three different bin numbers, in row 2, the respective mean velocities in these bins are shown and in row 3, the downstream evolution of the mean local TI calculated for a bin number of 500 is shown as interpolated contour plot. The red + marks the measurement position of the gust from row 1 and 2.

*in the wind tunnel as compared to $\approx 10\ s/1\ m$ for full scale measurements."*

We choose theses cases because they are separated by one order of magnitude and because they represent maybe the lowest and the highest frequency magnitude we will perform with this system. We agree that a more detailed investigation would be interesting. However, from this investigation, we could show that the underlying mechanisms that generate the perturbation are at least partially universal in the frequency range (cf. section 3.3 and conclusions). In addition, we saw that the blockage of the chopper blade needs to be reduced. This implies a new design of the chopper blade including a new characterization. Before redoing this heavy characterization work, we wanted to summarize our first results and share it with the wind energy community. The feedback we received will help us to improve our experimental investigations in the future.

**MINOR CORRECTIONS:**

1) (ln. 26) "extend" should be "extent". 2) (paragraph starting on ln. 30) The Wester et al. (2018) experiment is described as using the same configuration as Wei et al. (2019b), but the Wester et al. experiment actually preceded the Wei et al. experiment. Perhaps the words used or the order they are presented in should be changed.

**Answer**

Thank you, the two remarks have been addressed.

3) (ln. 38) The Makita paper should be referenced by the last name.

**Answer**

Thank you, this was addressed.

4) There are quite a few times in the manuscript (e.g., the sentence starting on line 55) where a preposition, e.g., "in", is missing after "downstream" or "upstream". These should be found and corrected.

**Answer**

Thank you, this was corrected throughout the text.

5) (ln. 99) Use of word "over" here is confusing. That terminology most often is used to mean "divided by" but that isn't what the authors mean.

**Answer**

Thank you, this has been corrected.    6) (ln. 99) Word "of" is missing in front of "the chopper".

**Answer**

Thank you, this has been corrected.

7) (ln. 124 and ln. 152) "the to the flow. . ." is some sort of typographical error that occurs more than once.

**Answer**

Thank you, this was corrected throughout the text (German sentence structures were accidentally used)

8) (ln. 154) "a low chopper frequency of the chopper", correct this.

**Answer**

Thank you, this has been corrected.

9) (ln. 201) Repeated "into".

**Answer**

Thank you, this has been corrected.

10) In general, the structure and writing style are good, but there are a few typos and missing articles and prepositions that confuse things. Please review the writing in general and perhaps get another set of eyes.

**Answer**

Thank you, we carefully revised the document.

[revised manuscript text omitted]
 10\,\text{s}$ and $\Delta t(f_{CH2}) \approx 0.9\,\text{s}$. $\Delta t(f_{CH1}) \approx 10\,\text{s}$ matches the characteristic time of the IEC
325 EOG $\Delta t_{IEC} = 10.5\,\text{s}$ closely. $\Delta t(f_{CH2}) \approx 0.9\,\text{s}$ is scaled to wind tunnel dimensions: The ratio of chord length $c$ and gust characteristic time should be constant which is achieved for the airfoil that we will use with $c = 0.09\,\text{m}$.

**3.4 Gust fluctuations: $u'$**

After investigating the mean gust properties and the underlying inverse gust shape, next, some turbulence characteristics of the
330 gust will be discussed by investigating the gust fluctuations $u'$. Figure 15 shows the energy spectral density of the turbulent fluctuations $u'$ (cf. figure 11 (d)) of all individual inverse gusts captured in the above-discussed time series measured at $X/w_{CH} = 5.425$ for the respective chopper frequencies. In addition, the average over all spectra is plotted in red, and the decay according to $E(f) \propto f^{-5/3}$ in the inertial sub-range is indicated in blue. The plots show that the turbulence within the inverse gust has approximately the same energy spectrum for all inverse gusts. The inertial sub-range follows a decay according to $E(f) \propto f^{-5/3}$
335 in the frequency range between $20\,\text{Hz} \leq f \leq 2000\,\text{Hz}$ ($f_{CH1} = 0.04\,\text{Hz}$) and $20\,\text{Hz} \leq f \leq 5000\,\text{Hz}$ ($f_{CH2} = 0.4\,\text{Hz}$), respec-

[revised manuscript text omitted]

---

## Author Response (AR2)

**Answers to the editors' comments received for the article**
**First characterization of a new perturbation system for gust generation: The Chopper**

Ingrid Neunaber and Caroline Braud

May 18, 2020

We would like to thank the editor and the associated editor for their comments and help. We did address the points in the following manner:

———

**» Line 13-15 – a bit confusing, what do you mean by performances? And why unmanned aerial vehicle research as an example?**

We agree that this sentence was confusing and we decided to delete the sentence as it does not add necessary information. Initially, the example of unmanned aerial vehicle research was given as transverse gusts are often researched with respect to drones.

———

**»Line 19 – accounted for is strange wording, expand on what you mean. Also the issue is not just that control mechanisms can not react quickly enough but that there isn't visibility of the encroaching gust (control engineers use the metaphor of a blind-folded boxer to describe the lack of awareness the turbine has óf the wind inflow)**

Thank you for pointing this out, we changed the phrase, „ As neither the inflow nor the flow around the blades are normally known, today's control mechanisms are incapable of compensating the impact of gusts which causes high loads.“ and also placed it in ll. 26 to improve the readability of the text.

———

**» Line 21 – IEC is a standard not a norm**

Thank you, we changed it throughout the document.

———

**» Line 73 – the device presented here**

Thank you, this was changed.

———

**» Line 80-81 – odd sentence structure, recommend rewording. Use another word than targeted as you use it in the previous sentence but with slightly different intent**

We agree that this sentence was not clear and rephrased the part of the paragraph (ll. 75): „The objective of this new perturbation system, called the "chopper", is to combine sudden, strong mean flow changes and turbulence which is targeted separately in other perturbation devices. This setup will help to improve the insight into blade and rotor aerodynamics under turbulent and unsteady conditions. This is important as the global pitch control can not mitigate the loads caused by unsteady aerodynamic effects due to gusts.“

———

**» Line 92 – recommendations (guidelines has a bit of a different meaning)**

Thank you, we changed this.

In addition to addressing the specific comments, we also made minor changes in the text to improve the readability. Those changes to not alter any information given in the paper. In the following, the changes in the document are marked:

**First characterization of a new perturbation system for gust generation: The Chopper**

Ingrid Neunaber[1] and Caroline Braud[1]

[1]LHEEA - Ecole Centrale Nantes, CNRS, 1 Rue de la Noë, 44321 Nantes, France

**Correspondence:** Ingrid Neunaber (ingrid.neunaber@ec-nantes.fr)

**Abstract.** We present a new system for the generation of rapid, strong flow  perturbations in the aerodynamic wind tunnel at Ecole Centrale Nantes. The system is called the *chopper*, and it consists of a rotating bar cutting through the inlet of a wind tunnel test section, thus generating an inverse gust that travels downstream. The flow generated by the chopper is investigated with respect to the rotational frequency using an array equipped with hot-wires that is traversed downstream in the flow field. It is found that the gust can be described as a superposition of the mean gust velocity, an underlying gust shape, and additional turbulence. Following this approach, the evolution of the mean gust velocity and turbulence intensity are presented, and the evolution of the underlying inverse gust shape is explained. The turbulence is shown to be characterized by an integral length scale of approximately half the chopper blade width and a turbulence decay according to $E(f) \propto f^{-5/3}$.

**1 Introduction**

The atmospheric boundary layer is naturally turbulent, and these turbulence conditions are highly complex and non-stationary. One characteristic of atmospheric turbulence is  that compared to a Gaussian distribution, the probability of extreme events, i.e. gusts,  is significantly higher, and this effect is called *intermittency* (see e.g. Morales et al. (2011)).  While the term "gust" is most commonly used for a strong change of the wind velocity in stream-wise direction (see e.g. Bardal and Sætran (2016) and Letson et al. (2019)),  gusts can also occur in transverse direction, for example in form of thunderstorm downbursts (see e.g. Nguyen and Manuel (2014)), or in form of sudden wind direction changes. This shows that the working conditions in  the lower part of the  atmospheric boundary layer are challenging.

Wind  turbines are operating in this part of the atmospheric boundary layer, and one consequence  is that they experience high loads, partially due to the intermittency in the wind (see e.g. Frandsen (2007), Lee et al. (2012) and Schwarz et al. (2019)).  To still ensure the durability and safety of a wind turbine during its planned lifetime of generally 20 years, the IEC-61400-1  standard gives a set of design requirements (cf. IEC-61400-1-4 (2019)) that include the turbulence characteristics and the occurrence of extreme wind conditions.

One  extreme wind condition is the extreme operating gust (EOG), an artificial gust that is assumed to occur once  in 50 years. It is modeled as a Mexican hat wavelet and characterized by a fixed characteristic time $\Delta t_{IEC} = 10.5\,\text{s}$, a rise and fall time $t_r = t_f = 2.8\,\text{s}$, and a velocity amplitude $\Delta u$[1]. $\Delta u$ can be in the order of magnitude of $10\,\text{ms}^{-1}$ for an inflow velocity close to the turbine's cut-off velocity[2]. One particular consequence of these strong and rapid gusts is that the aerodynamics at the rotor blade are  heavily affected, and the interactions between the three-dimensional, turbulent inflow and the rotor blades are highly complex and not well understood. As neither the inflow nor the flow around the blades are normally known, today's control mechanisms are incapable of compensating the impact of gusts which causes high loads.

To investigate the  aerodynamic reaction of rotor blades to turbulence and gusts, laboratory scale experimental research is an acknowledged method. Here, the inclusion of turbulence and unsteady  flows is important to model the real atmospheric conditions more closely. Over the years, different approaches have been used to include turbulence in wind tunnel experiments, and they can be classified into two categories: passive and active devices. An example of passive devices are grids. They are often used in the wind tunnel to include turbulence in airfoil and rotor aerodynamics investigations, as  e.g. done by Devinant et al. (2002), Sicot et al. (2006) and Sicot et al. (2008) for airfoils and Bartl et al. (2018) for  wakes of a lab scale turbine. For the investigation of  model wind turbines, another passive approach is to model the atmospheric boundary layer by means of static elements, as discussed e.g. by Lee et al. (2004) and Hancock and Hayden (2018) and used for example by Aubrun et al. (2013), Bastankhah and Porté-Agel (2017) and Chamorro et al. (2012). Here,  devices such as spikes, crossbars, and/or roughness elements are used to produce mean velocity and turbulence profiles of neutral or weakly stable atmospheric flows  by a long series of trial and error tests. While these setups give important insight into the impact of turbulence, they are limited to stationary investigations. Also, as for example Bartl et al. (2018) write, "the inevitable decay of the grid-generated turbulence in the experiment is not representative for real conditions".

To overcome these limitations, setups that actively control the inflow in the wind tunnel can be used.

One example  is the generation of transverse gusts by inducing a flow perpendicular to the  stream-wise direction, as done for example by Corkery et al. in a water channel or Poudel et al. (2018) in a wind tunnel to investigate the response of a flat plate respectively airfoil to a vertical flow. While in the former two examples, an inlet and an outlet exist for the transverse gust to pass through, Zhang et al. (2015) simulate a thunderstorm downburst similarly by blowing air down onto the test section floor.

To generate stream-wise gusts in the test section, the flow can  be altered globally in the wind tunnel. Horlock (1974) use flexible test section walls in a "wavy-wall" wind tunnel that can be driven to produce sinusoidal transverse gusts and stream-wise gusts. In Sarkar and Haan, jr. (2008), a system of bypass ducts is installed in a closed-loop wind tunnel to change
* * *
[1]The velocity amplitude $\Delta u$ depends on the velocity $u$ at hub height $z_{hub}$, the rotor diameter $D$ and the turbine class.

[2]For example, the EOG used for a wind turbine of class I.A with diameter $D = 136\,\text{m}$ and hub height $z_{hub} = 155\,\text{m}$ has an amplitude of $\Delta u = 9.8\,\text{ms}^{-1}$ at $u = 25\,\text{ms}^{-1}$.

the flow, and they achieve a 27% velocity increase in $2.2\,$s. Greenblatt (2016) uses a blockage modulation at the end of the test section of an open jet facility to induce flow variations. As these approaches focus on the mean flow variation, turbulence is

60 not added to the gusts.

Another approach to generate stream-wise gusts is to alter the flow directly at the beginning of the test section with active devices. The methods depend on the aimed gust shape. One possibility is the generation of periodic gusts in the wind tunnel. Tang et al. (1996) use rotating slotted cylinders to generate periodic gusts in the lateral and longitudinal directions, and they are able to create sinusoidal velocity fluctuations with an amplitude of $\Delta u = 5\,\text{ms}^{-1}$ at an inflow velocity of $u = 23\,\text{ms}^{-1}$.

65 While the velocity amplitude is only half of the presented EOG's velocity amplitude at $u = 25\,\text{ms}^{-1}$, one advantage of this system is the simple, low-cost design. Another method is to use a single pitching and plunging airfoil, as Wei et al. (2019a) demonstrate, who aim for a low-turbulent, lateral, periodic gust. Wester et al. (2018) and Wei et al. (2019b) use an array of oscillating 2D airfoils at the inlet of the wind tunnel. Wester et al. (2018) generate a lateral gust similar to the artificial EOG proposed in the IEC-61400-1  standard while Wei et al. (2019b) intend to reproduce an "ideal" sinusoidal gust without

70 turbulence in lateral and longitudinal direction.

Another device that is capable of creating customized flows and also gusts is an active grid. The most common design was proposed by Makita (1991), the so-called 'Makita-style' active grid, initially to generate homogeneous, quasi-isotropic turbulence in small wind tunnels. This initial design has been modified by different groups over the past years (e.g. Bodenschatz et al. (2014) and Hearst and Lavoie (2015)), and different control strategies have been applied - not only to push turbulence

75 research but also to use customizable flows for wind tunnel experiments. Cekli and van de Water (2010) and Hearst and Ganapathisubramani (2017) generate sheared flow profiles. Wind fields with atmospheric-like statistics are generated in Knebel et al. (2011). Traphan et al. (2018) and Petrović et al. (2019) demonstrate that Makita-style active grids can be used to generate gusts matching the one proposed as EOG in IEC-61400-1-4 (2019). While the flexibility and the stage of research are two major advantages of the Makita-style active grid, disadvantages include high complexity, high cost and maintenance. Also,

80  active grids struggle to create sudden, strong flow changes such as those   produced with the device presented here.

Overall, there are different systems to generate gusts and specific flows   . Many of the above-presented experiments focus on the generation of  gusts and flows tailored to investigate

85 isolated aerodynamic effects.

In this study, we will present a new perturbation system with a unique mechanism consisting of a rotating bar. The objective of this new perturbation system, called the "chopper", is to combine sudden, strong mean flow changes and   turbulence which is targeted separately in other perturbation devices.

90 This setup will  help to improve the insight into blade and rotor aerodynamics   under turbulent and unsteady conditions. This is important as the global pitch control can not mitigate the loads caused by unsteady aerodynamic effects due to gusts. On the contrary, active

flow control (AFC) strategies using sensors and actuators that are placed locally at the blades may be able to do so. Using for example plasma actuators, a fast and local control with up to $10\,\text{kHz}$ can be achieved, and up to a few kHz can be achieved in case of micro-jet actuators (see e.g. Leroy et al. (2016) and Jaunet and Braud (2018)). Those local flow control  systems are suitable to alleviate loads from rapid and turbulent atmospheric perturbations and thus decrease fatigue loads. Therefore, another particularly interesting application of this setup is the development of AFC devices like the actuators mentioned above or sensors, for example the e-TellTale sensor Soulier et al. (2020).

The present paper is a proof of concept of this new system, and it focuses on investigating the mean flow behavior, the shape and the characteristic time of the perturbation, and the turbulence generated by the system throughout the test section. The amplitude and the characteristic time of the perturbation will be compared to the IEC EOG. Further customization of the perturbation system will be needed to match realistic gusts like the one discussed for example by Bardal and Sætran (2016) which are far from the ideal gust shape provided by the IEC-61400-1 standard. Therefore, another objective of this proof-of-concept article is to propose some  recommendations towards that direction. We would like to note that the adaptation of perturbation systems and the flows they produce is done over long periods of time which was demonstrated by the progress in active grid research and the customization of atmospheric flows in wind tunnel facilities.

The paper is organized as follows: In section 2, the experimental setup will be introduced including a description of the chopper, section 3 will present results including the analysis of the time series (section 3.1), the mean gust flow field (section 3.2), the underlying shape (section 3.3) and the gust fluctuations (section 3.4), and section 4 will conclude the paper with an outlook.

**2   Experimental setup**

In the following, the chopper and the experimental setup used to characterize the flow disturbance generated by  it are introduced. In figure 1, a sketch of the chopper and its installation in the closed-loop wind tunnel can be found.  The chopper consists of a straight rectangular bar, the *chopper blade*, that rotates around its central axis which is aligned parallel to the test section. Within one revolution, the two arms of the chopper blade cross the inlet of the test section.  Where the chopper blade crosses the inlet, the gap acts as a discontinuity in the test section. The setup 
[revised manuscript text omitted]
 10\text{s}$ and $\Delta t(f_{CH2}) \approx 0.9\text{s}$. $\Delta t(f_{CH1}) \approx 10\text{s}$ matches the characteristic time of the IEC EOG $\Delta t_{IEC} = 10.5\text{s}$ closely. $\Delta t(f_{CH2}) \approx 0.9\,\text{s}$ is scaled to wind tunnel dimensions: The ratio of chord length $c$ and gust characteristic time should be constant which is achieved for the airfoil that we will use with $c = 0.09\,\text{m}$.

315 ## 3.4 Gust fluctuations: $u'$

After investigating the mean gust properties and the underlying inverse gust shape, next, some turbulence characteristics of the gust will be discussed by investigating the gust fluctuations $u'$. Figure 15 shows the energy spectral density of the turbulent fluctuations $u'$ (cf. figure 11 (d)) of all individual inverse gusts captured in the above-discussed time series measured at $X/w_{CH} = 5.425$ for the respective chopper frequencies. In addition, the average over all spectra is plotted in red, and the
320 decay according to $E(f) \propto f^{-5/3}$ in the inertial sub-range is indicated in blue. The plots show that the turbulence within the inverse gust has approximately the same energy spectrum for all inverse gusts. The inertial sub-range follows a decay according to $E(f) \propto f^{-5/3}$ in the frequency range between $20\,\text{Hz} \leq f \leq 2000\,\text{Hz}$ ($f_{CH1} = 0.04\,\text{Hz}$) and $20\,\text{Hz} \leq f \leq 5000\,\text{Hz}$ ($f_{CH2} = 0.4\,\text{Hz}$), respectively. As above-mentioned, we do not expect Kolmogorov's theory to hold in this flow which leads to

[revised manuscript text omitted]